# Personalized Bayesian Federated Learning with Wasserstein Barycenter Aggregation

**Ting Wei**
School of Statistics,
Renmin University of China
Beijing, China
weiting1006@ruc.edu.cn

**Biao Mei**
School of Statistics,
Renmin University of China
Beijing, China
2022103705@ruc.edu.cn

**Junliang Lyu**
Guanghua School of Management,
Peking University
Beijing, China
lvjunliang0211@ruc.edu.cn

**Renquan Zhang**
School of Mathematics Science,
Dalian University of Technology
Dalian, China
zhangrenquan@dlut.edu.cn

**Feng Zhou**[*]
Center for Applied Statistics
and School of Statistics,
Renmin University of China
Beijing Advanced Innovation Center for
Future Blockchain and Privacy Computing
Beijing, China
feng.zhou@ruc.edu.cn

**Yifan Sun**[*]
Center for Applied Statistics
and School of Statistics,
Renmin University of China
Beijing Advanced Innovation Center for
Future Blockchain and Privacy Computing
Beijing, China
sunyifan@ruc.edu.cn

## Abstract

Personalized Bayesian federated learning (PBFL) handles non-i.i.d. client data and quantifies uncertainty by combining personalization with Bayesian inference. However, existing PBFL methods face two limitations: restrictive parametric assumptions in client posterior inference and naive parameter averaging for server aggregation. To overcome these issues, we propose FedWBA, a novel PBFL method that enhances both local inference and global aggregation. At the client level, we use particle-based variational inference for nonparametric posterior representation. At the server level, we introduce particle-based Wasserstein barycenter aggregation, offering a more geometrically meaningful approach. Theoretically, we provide local and global convergence guarantees for FedWBA. Locally, we prove a KL divergence decrease lower bound per iteration for variational inference convergence. Globally, we show that the Wasserstein barycenter converges to the true parameter as the client data size increases. Empirically, experiments show that FedWBA outperforms baselines in prediction accuracy, uncertainty calibration, and convergence rate, with ablation studies confirming its robustness.

## 1 Introduction

Federated learning (FL) enables privacy-preserving collaborative model training across decentralized clients [43], making it particularly advantageous in privacy-sensitive domains such as finance [30],

---

[*]Corresponding authors.

39th Conference on Neural Information Processing Systems (NeurIPS 2025).

healthcare [35], and IoT [32]. Conventional FL, however, faces two key challenges from non-i.i.d client data: (1) convergence degradation due to client drift and (2) suboptimal global model performance [26, 40]. These limitations impede deployment in safety-critical fields. Personalized FL (PFL) addresses these limitations through client-specific adaptation techniques including transfer learning [13, 12], meta-learning [20, 14], and parameter decoupling [5, 10]. Nevertheless, conventional PFL frameworks relying on frequentist approaches remain vulnerable to overfitting and lack uncertainty quantification - crucial shortcomings for safety-critical applications. This motivates personalized Bayesian FL (PBFL) [11], which integrates Bayesian inference with PFL via two key mechanisms: prior regularization against overfitting and posterior inference for uncertainty quantification [45, 28].

Although PBFL has many advantages, existing methods face two main challenges: **(1)** Performing posterior inference on clients is challenging due to the lack of analytical solutions for model parameter posteriors, requiring approximation methods such as variational inference (VI) [9]. To simplify inference, many studies assume a parameterized variational distribution, such as Gaussian, to obtain a tractable evidence lower bound (ELBO) [45, 33, 46]. However, this parameterization introduces errors, as the true posterior is often complex and non-Gaussian [8]. **(2)** Performing aggregation on the central server is challenging because aggregating the posteriors from clients is not straightforward. To simplify the aggregation, many works upload the parameters of variational distributions from the clients and then average these parameters on the server to obtain the updated global prior [45, 7]. However, this is problematic since information geometry [4] indicates that distributions lie on a manifold, where "averaging" differs from that in parameter space.

To address the aforementioned issues, we propose a novel PBFL method, federated learning with Wasserstein barycenter aggregation (**FedWBA**), which enhances both local posterior inference and global aggregation. *At the local level*, we avoid parameterizing the variational distribution and instead employ particle-based VI. It represents the posterior with a set of particles, offering more nonparametric flexibility than parameterized VI. *At the global level*, we propose a particle-based Wasserstein barycenter aggregation method. It finds the barycenter of local posteriors on a manifold induced by the Wasserstein distance, combining client particles into a new set for the global prior. This method has a clearer geometric interpretation than parameter averaging. *Theoretically*, we guarantee the convergence of the proposed method. Our main contributions are as follows:

**(1)** We propose a novel PBFL method called FedWBA. At the local level, we use particle-based VI for greater nonparametric flexibility, while at the global level, we introduce a particle-based Wasserstein barycenter aggregation, which is more geometrically meaningful.

**(2)** We provide theoretical convergence guarantees at both the local and global levels. Locally, we prove a lower bound on the Kullback-Leibler (KL) divergence decrease per iteration, ensuring the convergence of variational inference. Globally, we show that as client data size approaches infinity, the Wasserstein barycenter converges to the true parameter.

**(3)** Comprehensive experiments demonstrate the superiority of FedWBA in prediction accuracy, uncertainty calibration, and convergence rate compared to baselines. Additionally, ablation studies evaluate the robustness of our approach w.r.t. different components.

## 2   Related Works

**Personalized Federated Learning** can be taxonomized into two principal paradigms according to [38]: The first personalizes global models through client-specific fine-tuning. Regularization-based methods, such as FedProx [25] add a proximal term to the loss function to measure model discrepancies, providing an intuitive and effective solution. Meta-learning approaches, such as Per-FedAvg [14] learn model initializations for rapid client adaptation, later enhanced by pFedMe [37] through Moreau envelope optimization. The second strategy focuses on personalizing the models by modifying the FL aggregation process, aligning with our PFL goal. Parameter-decoupling methods, exemplified by FedPer [5], decompose deep neural networks into shared base layers for feature extraction and client-specific heads for task adaptation. Multi-task learning methods, such as FedAMP [18] employs attention-based similarity weights, though sensitive to data quality. Our method can be considered as a form of the meta-learning approach.

**Bayesian Federated Learning** focuses on the inference and aggregation of posterior distributions over model parameters across clients. Depending on how the posterior is represented, BFL methods

can be broadly categorized into parametric and nonparametric approaches. Parametric methods assume a specific form for the posterior—typically a Gaussian—for tractability. Representative works include pFedBayes [45], FOLA [27], pFedVEM [46], FedPA [3], FedEP [17], and BA-BFL [19]. Although BA-BFL introduces geometric aggregation perspectives, it still relies on restrictive Gaussian posterior assumptions. FedHB [23] extends this to Gaussian mixtures, yet remains constrained by parametric forms. In contrast, nonparametric methods avoid strong assumptions on the posterior, enabling more flexible representations. For instance, FedPPD [7] employs MCMC to characterize local posteriors, while distributed SVGD (DSVGD) [22] adopts a particle-based approach. However, DSVGD requires three SVGD updates per communication round, which significantly increases computational cost. Our method advances this line of work by enabling single-round particle transport while preserving flexible posterior representations free from Gaussian constraints.

## 3 Preliminaries

In this section, we provide an overview of Stein variational gradient descent and Wasserstein barycenter aggregation.

### 3.1 Stein Variational Gradient Descent

SVGD approximates target distribution $p(\mathbf{x})$ by iteratively transforming particles from initial distribution $q(\mathbf{x})$ through $\mathbf{t}(\mathbf{x}) = \mathbf{x} + \epsilon\boldsymbol{\phi}(\mathbf{x})$, where $\epsilon$ is the step size and $\boldsymbol{\phi}(\cdot) : \mathbb{R}^M \to \mathbb{R}^M$ maximizes the KL divergence reduction rate. The objective is to find a $\mathbf{t}$ that minimizes $\mathrm{KL}(q_{[\mathbf{t}]}(\mathbf{x})\|p(\mathbf{x}))$. Computing the derivative of the KL w.r.t. $\epsilon$ at $\epsilon = 0$ yields a closed-form solution:

$$\nabla_\epsilon \mathrm{KL}(q_{[\mathbf{t}]}(\mathbf{x})\|p(\mathbf{x}))\big|_{\epsilon=0} = -\mathbb{E}_{q(\mathbf{x})}[\mathrm{trace}(\mathcal{A}_p\boldsymbol{\phi}(\mathbf{x}))],$$

where $\mathcal{A}_p\boldsymbol{\phi}(\mathbf{x}) = \nabla_\mathbf{x}\log p(\mathbf{x})\boldsymbol{\phi}(\mathbf{x})^\top + \nabla_\mathbf{x}\boldsymbol{\phi}(\mathbf{x})$. To maximize the rate of decrease in KL divergence, we aim to select $\boldsymbol{\phi}$ such that $\mathbb{E}_{q(\mathbf{x})}[\mathrm{trace}(\mathcal{A}_p\boldsymbol{\phi}(\mathbf{x}))]$ is as large as possible.

The constrained optimization is resolved via reproducing kernel Hilbert spaces (RKHS) [29]: Let $k(\cdot, \cdot)$ be a positive-definite kernel defining an RKHS $\mathcal{H}$, with $\mathcal{H}_D = \mathcal{H} \times \ldots \times \mathcal{H}$ denoting its D-dimensional product space for vector-valued functions $\mathbf{f} = (f_1, \ldots, f_D)$ where $f_i \in \mathcal{H}$. Constraining $\boldsymbol{\phi} \in \mathcal{H}_D$ with $\|\boldsymbol{\phi}\|_{\mathcal{H}_D} \leq 1$, the steepest descent direction admits the following analytic expression:

$$\boldsymbol{\phi}^*(\cdot) = \boldsymbol{\psi}(\cdot)/\|\boldsymbol{\psi}\|_{\mathcal{H}_D}, \quad \boldsymbol{\psi}(\cdot) = \mathbb{E}_{q(\mathbf{x})}[\mathcal{A}_p k(\mathbf{x}, \cdot)].$$

If we approximate the distribution $q(\mathbf{x})$ using a finite set of particles located at $\{\mathbf{x}_i\}_{i=1}^N$, the iterative algorithm at the $l$-th iteration can be formulated as follows:

$$\mathbf{x}_i^{(l+1)} = \mathbf{x}_i^{(l)} + \frac{\epsilon}{N}\sum_{j=1}^N \left[ k(\mathbf{x}_j^{(l)}, \mathbf{x}_i^{(l)})\nabla_\mathbf{x}\log p(\mathbf{x}) + \nabla_\mathbf{x}k(\mathbf{x}, \mathbf{x}_i^{(l)}) \right]\Big|_{\mathbf{x}=\mathbf{x}_j^{(l)}}.$$

The first term in the update formula attracts particles towards high-probability regions, while the second term acts as a repulsive force, preventing particle clustering and ensuring a more uniform exploration of the distribution's support.

### 3.2 Wasserstein Barycenter Aggregation

The Wasserstein distance from optimal transport theory provides geometric distribution comparison through minimal transport cost, with the 2-Wasserstein case defined as:

$$W_2^2(p(\mathbf{x}), q(\mathbf{x})) = \min_{\pi\in\Pi}\mathbb{E}_{\pi(\mathbf{x},\mathbf{x}')}[\|\mathbf{x} - \mathbf{x}'\|^2], \tag{1}$$

where $\pi(\mathbf{x}, \mathbf{x}')$ represents a joint distribution with marginals $p$ and $q$ respectively, and $\Pi$ denotes the set of all such joint distributions [39]. The Wasserstein distance has a clear physical interpretation. The norm term $\|\mathbf{x} - \mathbf{x}'\|^2$ represents the cost of transporting a unit mass from $\mathbf{x}$ to $\mathbf{x}'$, and $\pi(\mathbf{x}, \mathbf{x}')$ is a transport plan that specifies the amount of mass to move from $\mathbf{x}$ to $\mathbf{x}'$. Therefore, the Wasserstein distance is the minimal transportation cost, and the optimal $\pi^*(\mathbf{x}, \mathbf{x}')$ is the plan that achieves this minimal cost among all plans.

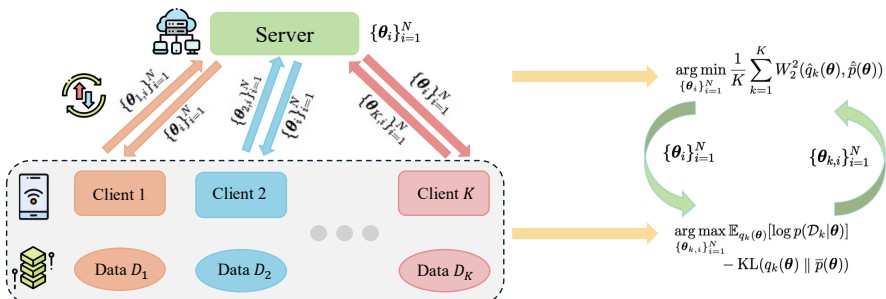

Figure 1: Overview of FedWBA. Left: System diagram. Clients upload local posterior particles to server for aggregation, server updates global prior particles and redistributes them to clients. Right: Local posterior particles from maximizing local ELBO, global prior particles as Wasserstein barycenter of $K$ local posteriors.

The Wasserstein barycenter provides a geometric mean of distributions by minimizing the weighted sum of Wasserstein distances to given distributions [2]. For a set of probability distributions $\{p_i\}_{i=1}^{K}$ with weights $\{w_i\}_{i=1}^{K}$ satisfying $\sum_{i=1}^{K} w_i = 1, w_i \geq 0$, the Wasserstein barycenter is defined as:

$$\overline{p} = \arg\min_{p \in \mathcal{P}} \frac{1}{K} \sum_{i=1}^{K} w_i W_2^2(p, p_i),$$

where $\mathcal{P}$ is the set of probability distributions. The Wasserstein barycenter has the advantage of a clear geometric interpretation, but a drawback is the challenging computation of the Wasserstein distance. This difficulty arises because identifying the optimal transport plan between continuous distributions is complex and analytically tractable only in special cases [2]. Nevertheless, it is worth noting that when the given distributions are discrete, the Wasserstein distance can be estimated via linear programming, making the computation of the barycenter tractable [6, 15].

## 4 Methodology

In this section, we introduce our federated learning with Wasserstein barycenter aggregation (Fed-WBA) approach. As illustrated in Figure 1, the training process is iterative: clients begin by downloading the global prior from the server. Using this global prior, each client updates its local posterior, represented as a set of particles, via SVGD. The updated particles are then uploaded to the server for aggregation, performed using Wasserstein barycenter aggregation. This iterative process continues until convergence. Each client adapts the global prior to its local data, ensuring personalization, while the global prior, formed by aggregating local posteriors, reflects overall patterns.

### 4.1 Problem Definition

We consider a distributed system consisting of a single server and $K$ clients. Assume all clients share the same model with parameters $\boldsymbol{\theta} \in \mathbb{R}^M$, and each of the $K$ clients has its own dataset $\{\mathcal{D}_k\}_{k=1}^{K}$. We represent the global prior on the server using a set of particles $\{\boldsymbol{\theta}_i\}_{i=1}^{N}$, and similarly, use $\{\boldsymbol{\theta}_{k,i}\}_{i=1}^{N}$ to represent the local posterior on the $k$-th client. Distributions without a hat denote continuous distributions, while those with a hat represent the corresponding empirical (discrete) distributions expressed through particles.

### 4.2 Local Posterior via Stein Variational Gradient Descent

In each communication round, on the $k$-th client, a global prior over the model parameters $\overline{p}(\boldsymbol{\theta})$ is downloaded from the server. Our goal is to adapt this prior to the local data using Bayes' rule, resulting in the posterior distribution of the model parameters:

$$p(\boldsymbol{\theta} \mid \mathcal{D}_k) \propto p(\mathcal{D}_k \mid \boldsymbol{\theta})\overline{p}(\boldsymbol{\theta}),$$

where $\overline{p}(\boldsymbol{\theta})$ is the prior, $p(\mathcal{D}_k \mid \boldsymbol{\theta})$ is the categorical likelihood, and $p(\boldsymbol{\theta} \mid \mathcal{D}_k)$ is the posterior. Since the likelihood is usually parameterized by a neural network, it is typically non-conjugate to the prior.

As a result, the posterior generally lacks an analytical expression. To address this, approximate inference methods are employed to approximate the posterior. VI is among the most widely used approaches for this purpose. Specifically, in VI, the posterior $p(\boldsymbol{\theta} \mid \mathcal{D}_k)$ is approximated by a variational distribution $q_k(\boldsymbol{\theta})$. The optimal variational distribution is obtained by minimizing the KL divergence between them or, equivalently, by maximizing the ELBO:

$$F(q_k(\boldsymbol{\theta})) = \mathbb{E}_{q_k(\boldsymbol{\theta})}[\log p(\mathcal{D}_k|\boldsymbol{\theta})] - \mathrm{KL}\big(q_k(\boldsymbol{\theta}) \parallel \overline{p}(\boldsymbol{\theta})\big).$$

Prior methods [45, 33, 46] commonly impose parametric assumptions (e.g., Gaussian) on $q_k(\boldsymbol{\theta})$ for ELBO tractability, yet suffer from approximation mismatch when the true posterior deviates from assumed forms [8]. Inspired by [22], we advocate for using the SVGD method, which employs a set of particles to flexibly represent the variational distribution. This approach eliminates the need for specific parametric assumptions. Specifically, we assume that the variational distribution $q_k(\boldsymbol{\theta})$ is represented by a set of particles $\{\boldsymbol{\theta}_{k,i}\}_{i=1}^N$, which are iteratively updated according to SVGD. The particle update rule in the $(l)$-th iteration is given by:

$$\boldsymbol{\theta}_{k,i}^{(l+1)} = \boldsymbol{\theta}_{k,i}^{(l)} + \frac{\epsilon}{N} \sum_{j=1}^N \left[ k(\boldsymbol{\theta}_{k,j}^{(l)}, \boldsymbol{\theta}_{k,i}^{(l)}) \nabla_{\boldsymbol{\theta}} \log p(\boldsymbol{\theta} \mid \mathcal{D}_k) + \nabla_{\boldsymbol{\theta}} k(\boldsymbol{\theta}, \boldsymbol{\theta}_{k,i}^{(l)}) \right] \Big|_{\boldsymbol{\theta} = \boldsymbol{\theta}_{k,j}^{(l)}}, \qquad (2)$$

where $\nabla_{\boldsymbol{\theta}} \log p(\boldsymbol{\theta} \mid \mathcal{D}_k) = \nabla_{\boldsymbol{\theta}} \log \overline{p}(\boldsymbol{\theta}) + \nabla_{\boldsymbol{\theta}} \log p(\mathcal{D}_k \mid \boldsymbol{\theta})$ according to Bayes' rule, $k(\cdot, \cdot)$ is a positive definite kernel corresponding to a RKHS.

As the number of particles $N$ increases, the empirical distribution of $\{\boldsymbol{\theta}_{k,i}\}_{i=1}^N$ asymptotically converges to the optimal variational distribution in the corresponding RKHS; when we use only a single particle $N = 1$, the maximum a posteriori (MAP) estimate can be obtained [29].

### 4.3 Global Prior via Wasserstein Barycenter Aggregation

In each communication round, the server needs to aggregate the posterior distributions uploaded by multiple clients into a global prior, which is a challenging procedure. Previous works [45, 7] used a naive approach by averaging the parameters of the uploaded posteriors and treating the resulting distribution as the aggregated one. We argue that this aggregation method is problematic because, according to information geometry [4], distributions lie on a manifold, where the "average" on the manifold differs from the "average" in the parameter space. A visual illustration of this difference can be seen in Figure 4 (Appendix A).

To address this issue, we propose using Wasserstein barycenter aggregation, leveraging the geometric properties of the manifold where the local posteriors reside. Unlike previous works, our approach replaces parameter averaging with Wasserstein distance-based averaging, offering a clearer geometric interpretation. Specifically, after applying SVGD, the variational distribution $q_k(\boldsymbol{\theta})$ on the $k$-th client is approximated as an empirical measure $\hat{q}_k(\boldsymbol{\theta})$ using particles $\{\boldsymbol{\theta}_{k,i}\}_{i=1}^N$; similarly, the global prior $\overline{p}(\boldsymbol{\theta})$ is also approximated as an empirical measure $\hat{\overline{p}}(\boldsymbol{\theta})$ using particles $\{\boldsymbol{\theta}_i\}_{i=1}^N$:

$$\hat{q}_k(\boldsymbol{\theta}) = \frac{1}{N} \sum_{i=1}^N \delta_{\boldsymbol{\theta}_{k,i}}(\boldsymbol{\theta}), \quad \hat{\overline{p}}(\boldsymbol{\theta}) = \frac{1}{N} \sum_{i=1}^N \delta_{\boldsymbol{\theta}_i}(\boldsymbol{\theta}),$$

where $\delta_{\boldsymbol{\theta}}(\cdot)$ is the Dirac measure. To define the Wasserstein distance between $\hat{\overline{p}}(\boldsymbol{\theta})$ and $\hat{q}_k(\boldsymbol{\theta})$, we define a distance matrix $\mathbf{M}_k \in \mathbb{R}^{N \times N}$ between the global prior particles and local posterior particles:

$$\mathbf{M}_{k,i,j} = \|\boldsymbol{\theta}_i - \boldsymbol{\theta}_{k,j}\|^2,$$

where $i, j$ represent the entry in the $i$-th row and $j$-th column. We then define the transport plan as a matrix $\mathbf{T}_k \in \mathbb{R}^{N \times N}$ which is a discrete version of the joint distribution $\pi(\cdot, \cdot)$ in Equation (1). Then the Wasserstein distance between $\hat{\overline{p}}(\boldsymbol{\theta})$ and $\hat{q}_k(\boldsymbol{\theta})$ can be written as:

$$W_2^2(\hat{\overline{p}}, \hat{q}_k) = \min_{\mathbf{T}_k \in \mathcal{T}} \langle \mathbf{M}_k, \mathbf{T}_k \rangle_F, \quad \text{s.t. } \mathbf{T}_k \mathbf{1} = \frac{1}{N}\mathbf{1}, \ \mathbf{T}_k^\top \mathbf{1} = \frac{1}{N}\mathbf{1}, \ \mathbf{T}_k \geq 0, \qquad (3)$$

where $\mathcal{T}$ is the set of all possible $\mathbf{T}_k$ and $\langle \cdot, \cdot \rangle_F$ is the Frobenius inner product between two matrices. The constraints of $\mathbf{T}_k$ correspond to the marginal properties of $\pi(\cdot, \cdot)$. Let $\mathbf{T}_k^*$ denote the optimal transportation plan in Equation (3). It is easy to see that this is a linear programming problem, which

can be solved using general optimization packages. However, more specialized algorithms, such as Orlin's algorithm, also exist [34, 15].

After obtaining $\mathbf{T}_k^*$, we can update the global prior as the Wasserstein barycenter of $K$ local posteriors:

$$\hat{\bar{p}}^*(\boldsymbol{\theta}) = \arg\min_{\{\boldsymbol{\theta}_i\}_{i=1}^N} \frac{1}{K} \sum_{k=1}^K \langle \mathbf{M}_k, \mathbf{T}_k^* \rangle_F . \tag{4}$$

Denoting $\boldsymbol{\Theta} = [\boldsymbol{\theta}_1, \ldots, \boldsymbol{\theta}_N]^\top \in \mathbb{R}^{N \times M}$ to be the stack of global prior particles and $\boldsymbol{\Theta}_k = [\boldsymbol{\theta}_{k,1}, \ldots, \boldsymbol{\theta}_{k,N}]^\top \in \mathbb{R}^{N \times M}$ to be the stack of local posterior particles on the $k$-th client, it is easy to prove that the objective function in Equation (4) is quadratic w.r.t. $\boldsymbol{\Theta}$, so the optimal $\boldsymbol{\Theta}^*$ has an analytical solution (see proof in Appendix B):

$$\boldsymbol{\Theta}^* = \frac{1}{K} \sum_{k=1}^K \mathbf{T}_k^* \boldsymbol{\Theta}_k \mathrm{diag}(N^{-1}). \tag{5}$$

After obtaining the optimal global prior particles $\{\boldsymbol{\theta}_i^*\}_{i=1}^N$, a continuous global prior is required on the clients for SVGD updates, as the gradient of the log-prior must be computed in Equation (2). To achieve this, we employ kernel density estimation (KDE) to construct a continuous global prior:

$$\bar{p}^*(\boldsymbol{\theta}) = \frac{1}{N} \sum_{i=1}^N \tilde{k}(\boldsymbol{\theta}, \boldsymbol{\theta}_i^*), \tag{6}$$

where $\tilde{k}(\cdot, \cdot)$ is a normalized kernel in KDE, which is different from the kernel $k(\cdot, \cdot)$ in SVGD.

### 4.4 Algorithm

In summary, at the client level, each client receives the global prior particles and reconstructs a continuous prior using Equation (6), then updates the local posterior particles via Equation (2). At the server level, the global prior particles are updated by aggregating the uploaded local posterior particles using Equation (5). We term this method FedWBA, with its pseudocode provided in Appendix C.

## 5 Theoretical Analysis

In this section, we aim to establish the convergence properties of our proposed method. *Locally*, we demonstrate that the ELBO increases with each iteration. By providing a lower bound for the difference in ELBO between consecutive iterations, we ensure that as the iterative process of SVGD proceeds, the variational distribution represented by particles is getting closer to the true posterior. *Globally*, we show that as the data size on each client tends to infinity, the Wasserstein barycenter converges to a delta measure centered at the true parameter. This convergence implies that the aggregated global prior effectively captures the overall patterns. The proofs are presented in Appendices D and E, respectively.

**Assumption 5.1.** $\nabla_{\boldsymbol{\theta}} \mathbf{t}(\boldsymbol{\theta})$ is a positive definite matrix, where the eigenvalues $e_i > 0$ for $i = 1, \cdots, D$.

**Assumption 5.2.** $\epsilon$ is small enough s.t. $\|\epsilon \nabla_{\boldsymbol{\theta}} \phi(\boldsymbol{\theta})\| < 1$.

**Theorem 5.3.** *Under Assumptions 5.1 and 5.2, given SVGD iteration $l$, with client $k$ scheduled, the increase in the ELBO from iteration $l$ to $l+1$ satisfies the inequality:*

$$F\big(q_k^{(l+1)}(\boldsymbol{\theta})\big) - F\big(q_k^{(l)}(\boldsymbol{\theta})\big) \geq \epsilon D\big(q_k^{(l)}(\boldsymbol{\theta}), \tilde{p}(\boldsymbol{\theta} \mid \mathcal{D}_k)\big), \tag{7}$$

*where $\epsilon$ is the step size in SVGD, $\tilde{p}(\boldsymbol{\theta} \mid \mathcal{D}_k)$ denotes the unnormalized posterior distribution and $D(q, p)$ stands for the kernelized Stein discrepancy:*

$$D(q, p) = \max_{\phi \in \mathcal{H}_D} \{\mathbb{E}_{q(\boldsymbol{\theta})}[trace(A_p \phi(\boldsymbol{\theta}))], \ s.t. \ \|\phi\|_{\mathcal{H}_D} \leq 1\}.$$

Theorem 5.3 shows that the ELBO increases with each iteration, which equivalently ensures that the KL divergence between the variational distribution and the true posterior decreases, with the lower bound of its decrease determined by the kernelized Stein discrepancy. This implies that $q_k(\boldsymbol{\theta})$ converges to $p(\boldsymbol{\theta} \mid \mathcal{D}_k)$ as the iterations progress.

**Assumption 5.4.** $\Xi$ is a compact space in $\rho$ metric, and $\boldsymbol{\theta}_0$ is an interior point of $\Xi$. All clients have equal data size, i.e., $s_k = S/K$ for $k = 1, \ldots, K$, where $s_k$ is the data size of client $k$ and $S$ is the total data size.

**Assumption 5.5.** For any $\boldsymbol{\theta}, \boldsymbol{\theta}' \in \Xi$ and $k = 1, \ldots, K$, there exist positive constants $\alpha$ and $C_L$, s.t. the following inequality holds:

$$h_{sk}^2(\boldsymbol{\theta}, \boldsymbol{\theta}') > C_L \rho^{2\alpha}(\boldsymbol{\theta}, \boldsymbol{\theta}').$$

**Assumption 5.6.** There exist constants $C_1 > 0$, $0 < C_2 < \frac{C_1^2}{2^{12}}$, a function $\Psi(u, r) > 0$ that is non-increasing in $u \in \mathbb{R}^+$ and non-decreasing in $r \in \mathbb{R}^+$. For all $k = 1, \ldots, K$, any $u, r > 0$, and all sufficiently large $s$, the generalized bracketed entropy $H_{[]}$ satisfies

$$H_{[]}\Big(u, \{p(D_k|\boldsymbol{\theta}) : \boldsymbol{\theta} \in \Xi, h_{sk}(\boldsymbol{\theta}, \boldsymbol{\theta}_0) \leq r\}, h_{sk}\Big) \leq \Psi(u, r) \quad \text{for all } k = 1, \ldots, K;$$

and

$$\int_{C_1 r^2/2^{12}}^{C_1 r} \sqrt{\Psi(u, r)} \, du < C_2 \sqrt{s} r^2.$$

**Assumption 5.7.** There exist positive constants $\kappa$ and $c_\pi$ s.t., uniformly over $k = 1, \ldots, K$,

$$\Pi\left(\boldsymbol{\theta} \in \Xi : \frac{1}{s} \sum_{i=1}^{s} \mathbb{E}_{P_{\boldsymbol{\theta}_0}} \exp\left(\kappa \log_+ \frac{p(D_{ki}|\boldsymbol{\theta}_0)}{p(D_{ki}|\boldsymbol{\theta})}\right) - 1 \leq \frac{\log^2 s}{s}\right) \geq \exp(-c_\pi K \log^2 s),$$

where $\log^+ x = \max(\log x, 0)$ for $x > 0$.

**Assumption 5.8.** The metric $\rho$ satisfies the following property: for any $N \in \mathbb{N}$, $\boldsymbol{\theta}_1, \ldots, \boldsymbol{\theta}_N, \boldsymbol{\theta}' \in \Xi$ and nonnegative weights $\sum_{i=1}^{N} w_i = 1$,

$$\rho\left(\sum_{i=1}^{N} w_i \boldsymbol{\theta}_i, \boldsymbol{\theta}'\right) \leq \sum_{i=1}^{N} w_i \rho(\boldsymbol{\theta}_i, \boldsymbol{\theta}').$$

**Theorem 5.9.** *Assuming Assumptions 5.4 to 5.8 hold for all client posteriors $\{p(\boldsymbol{\theta} \mid \mathcal{D}_k)\}_{k=1}^{K}$, let $s = |\mathcal{D}_k|$ denote the data size on each client. Then, as $s \to \infty$, we have:*

$$W_2(\overline{p}(\boldsymbol{\theta}), \delta_{\boldsymbol{\theta}_0}(\boldsymbol{\theta})) = O_p\left(\sqrt{\frac{\log^{2/\alpha} s}{s^{1/\alpha}}}\right),$$

*where $\delta_{\boldsymbol{\theta}_0}$ denotes the Dirac delta measure centered at the true parameter $\boldsymbol{\theta}_0$, the $O_p$ notation is w.r.t. the probability measure $P_{\boldsymbol{\theta}_0}^{(S)}$ and $\alpha$ is a positive constant introduced in Assumption 5.5.*

Theorem 5.9 shows that the Wasserstein barycenter converges to the true parameter $\boldsymbol{\theta}_0$ at a rate of $s^{-\frac{1}{2\alpha}}$ up to logarithmic factors, which demonstrates that our global aggregation mechanism effectively combines client information to approximate the true parameter.

## 6 Experiments

In this section, we utilize four real-world datasets to evaluate the performance of FedWBA in terms of prediction accuracy, uncertainty calibration, and convergence rate. We perform all experiments using a server with GPU (NVIDIA GeForce RTX 4090). [2]

### 6.1 Experimental Setup

**Baselines:** We conduct comprehensive comparisons against state-of-the-art FL/PFL methods implemented in the standardized framework [44]. Specifically, we compare our approach against the following FL methods: (1) **FedAvg** [31], (2) **FedProx** [25], (3) **Scaffold** [21]; PFL methods: (4) **FedPer** [5], (5) **PerFedAvg** [14], (6) **pFedME** [37]; and BFL methods : (7) **pFedBayes** [45], (8) **pFedVEM** [46], (9) **pFedGP** [1] and (10) **DSVGD** [22].

---

[2]Our code is publicly available at https://github.com/TingWei1006/FedWBA

Table 1: Test accuracy (% ± SEM) over 50, 100, 200 clients on MNIST, FMNIST, CIFAR-10 and CIFAR-100. Best and second-best results are **bolded** and underlined, respectively. Dashes denote DSVGD results unavailable within 10 hours on CIFAR-10/CIFAR-100.

| DATASET | METHOD | 50 CLIENTS | 100 CLIENTS | 200 CLIENTS | 50 CLIENTS | 100 CLIENTS | 200 CLIENTS | DATASET |
|---|---|---|---|---|---|---|---|---|
| MNIST | FEDAVG | $91.98 \pm 0.07$ | $91.76 \pm 0.08$ | $90.94 \pm 0.06$ | $59.24 \pm 0.70$ | $55.96 \pm 0.21$ | $51.78 \pm 0.36$ | CIFAR-10 |
| | FEDPROX | $92.12 \pm 0.08$ | $92.04 \pm 0.11$ | $90.82 \pm 0.16$ | $60.26 \pm 0.42$ | $58.87 \pm 0.60$ | $58.86 \pm 0.60$ | |
| | SCAFFOLD | $92.90 \pm 0.07$ | $92.14 \pm 0.08$ | $90.85 \pm 0.11$ | $62.90 \pm 0.38$ | $61.42 \pm 0.51$ | $60.51 \pm 0.60$ | |
| | FEDPER | $96.53 \pm 0.02$ | $95.98 \pm 0.05$ | $94.19 \pm 0.04$ | $\mathbf{73.53 \pm 0.07}$ | $68.95 \pm 0.21$ | $65.41 \pm 0.11$ | |
| | PERFEDAVG | $94.65 \pm 0.15$ | $93.60 \pm 0.12$ | $90.61 \pm 0.09$ | $67.70 \pm 0.41$ | $62.17 \pm 1.06$ | $61.70 \pm 0.90$ | |
| | PFEDME | $95.70 \pm 0.02$ | $95.60 \pm 0.02$ | $93.82 \pm 0.02$ | $72.62 \pm 0.17$ | $71.33 \pm 0.15$ | $69.72 \pm 0.22$ | |
| | PFEDBAYES | $95.83 \pm 0.05$ | $94.15 \pm 0.03$ | $92.77 \pm 0.10$ | $72.83 \pm 0.16$ | $68.62 \pm 0.18$ | $66.75 \pm 0.21$ | |
| | PFEDVEM | $97.90 \pm 0.05$ | $97.12 \pm 0.06$ | $96.42 \pm 0.10$ | $73.20 \pm 0.20$ | $71.90 \pm 0.1$ | $\mathbf{70.10 \pm 0.30}$ | |
| | PFEDGP | $97.69 \pm 0.15$ | $97.05 \pm 0.19$ | $96.48 \pm 0.21$ | $72.61 \pm 0.13$ | $71.87 \pm 0.22$ | $69.73 \pm 0.15$ | |
| | DSVGD | $96.41 \pm 0.02$ | $96.19 \pm 0.03$ | $95.97 \pm 0.01$ | – | – | – | |
| | **OURS** | $\mathbf{97.99 \pm 0.03}$ | $\mathbf{97.36 \pm 0.01}$ | $\mathbf{96.95 \pm 0.01}$ | $73.40 \pm 0.25$ | $\mathbf{72.68 \pm 0.14}$ | $69.83 \pm 0.03$ | |
| FMNIST | FEDAVG | $84.60 \pm 0.22$ | $84.06 \pm 0.09$ | $83.14 \pm 0.05$ | $25.09 \pm 0.14$ | $24.49 \pm 0.24$ | $21.53 \pm 0.19$ | CIFAR-100 |
| | FEDPROX | $84.65 \pm 0.08$ | $83.51 \pm 0.09$ | $82.95 \pm 0.15$ | $25.34 \pm 0.12$ | $34.56 \pm 0.31$ | $34.56 \pm 0.30$ | |
| | SCAFFOLD | $85.49 \pm 0.19$ | $83.57 \pm 0.10$ | $82.56 \pm 0.16$ | $47.45 \pm 0.67$ | $44.62 \pm 0.16$ | $41.48 \pm 0.50$ | |
| | FEDPER | $91.65 \pm 0.02$ | $90.01 \pm 0.07$ | $88.23 \pm 0.05$ | $52.88 \pm 0.22$ | $51.43 \pm 0.16$ | $38.88 \pm 0.29$ | |
| | PERFEDAVG | $90.03 \pm 0.10$ | $87.40 \pm 0.09$ | $85.03 \pm 0.03$ | $51.42 \pm 0.12$ | $49.02 \pm 0.29$ | $35.84 \pm 0.19$ | |
| | PFEDME | $90.32 \pm 0.14$ | $89.63 \pm 0.19$ | $89.19 \pm 0.23$ | $58.40 \pm 0.17$ | $57.66 \pm 0.19$ | $55.23 \pm 0.28$ | |
| | PFEDBAYES | $91.70 \pm 0.03$ | $91.53 \pm 0.06$ | $90.50 \pm 0.06$ | $61.45 \pm 0.10$ | $60.58 \pm 0.17$ | $55.92 \pm 0.34$ | |
| | PFEDVEM | $91.80 \pm 0.10$ | $91.40 \pm 0.10$ | $90.70 \pm 0.10$ | $57.57 \pm 0.41$ | $55.32 \pm 0.11$ | $49.95 \pm 0.35$ | |
| | PFEDGP | $91.83 \pm 0.05$ | $91.51 \pm 0.07$ | $90.58 \pm 0.15$ | $63.30 \pm 0.10$ | $61.30 \pm 0.20$ | $59.10 \pm 0.22$ | |
| | DSVGD | $92.41 \pm 0.04$ | $91.33 \pm 0.02$ | $90.67 \pm 0.02$ | – | – | – | |
| | **OURS** | $\mathbf{92.50 \pm 0.06}$ | $\mathbf{91.65 \pm 0.02}$ | $\mathbf{90.97 \pm 0.08}$ | $\mathbf{64.21 \pm 0.02}$ | $\mathbf{61.73 \pm 0.16}$ | $\mathbf{59.22 \pm 0.12}$ | |

**Datasets:** To benchmark under realistic non-i.i.d. conditions with label skew, we evaluate on four vision datasets: **MNIST**, FashionMNIST (**FMNIST**) [42], **CIFAR-10**, and **CIFAR-100** [24]. We adopt the setup from [45, 46, 1] in which each client receives 5 unique labels for MNIST, FMNIST, and CIFAR-10, and 10 labels sampled from distinct superclasses for CIFAR-100.

**Setup:** We conduct all experiments with 100 communication rounds and 20% client participation per round, sufficient for algorithm convergence. We evaluate performance with client counts $K \in \{50, 100, 200\}$, considering that more clients disperse the training data. Following established architectures in prior work, we implement lightweight models: for MNIST and FMNIST, adopting the single-hidden-layer MLP from [45, 46], and for CIFAR-10/100, deploying the LeNet-style CNN in [1] to accommodate resource constraints.

**Hyperparameter:** Similar to [29], in all SVGD experiments, we employ the radial basis function (RBF) kernel $k(\boldsymbol{\theta}, \boldsymbol{\theta}_0) = \exp(-\frac{\|\boldsymbol{\theta} - \boldsymbol{\theta}_0\|_2^2}{h})$. The bandwidth $h$ is set to $h = \mathrm{med}^2 / \log N$, with med being the median of pairwise particle distances in the current iteration. The bandwidth of the Gaussian kernel $\tilde{k}(\cdot, \cdot)$ used for KDE is set to 0.55. We use AdaGrad with momentum to set the learning rate $\epsilon$. Considering communication constraints from uploaded data size, we set the number of particles to 10, balancing computational accuracy and communication overhead in SVGD, mitigating communication bottlenecks without sacrificing much variational inference performance.

## 6.2 Performance of Prediction

As shown in Table 1, the algorithm performance is reported under optimal hyperparameter configurations. Given that the global model acts as a prior regularizer, we primarily focus on client-side performance; additionally, we designed experiments to verify that the global model's performance is comparable to FL baselines, as detailed in Appendix F.1. Due to DSVGD failing to produce results within 10 hours on CIFAR-10 and CIFAR-100, we only report its performance on MNIST and FMNIST. As is evident in Table 1, the proposed method outperforms the existing approaches in nearly all scenarios. This demonstrates FedWBA's strong adaptability to diverse real datasets, delivering consistent performance irrespective of data type or scale. Partial results are copied from [1, 46].

The performance superiority of FedWBA stems from its unique nonparametric design. In our approach, particles flexibly represent both the local posterior and the global prior, eliminating the need for parametric constraints. This nonparametric representation gives the model strong fitting ability, allowing it to better capture data patterns, thus having an edge over baseline methods with parametric assumptions. Also, as a Bayesian algorithm, FedWBA excels in few-shot scenarios. When

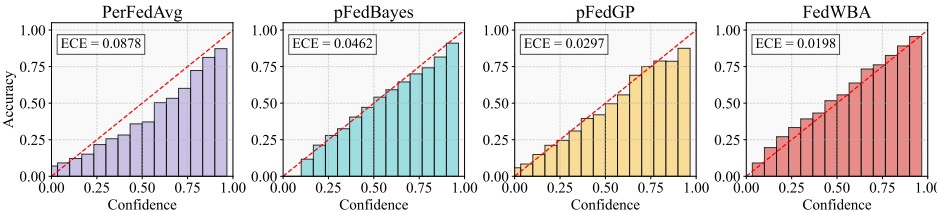

Figure 2: Reliability diagrams of top four methods on CIFAR-100. The perfect calibration is plotted as a red diagonal, and the actual results are presented as bar charts. The gap between the top of each bar and the red line represents the calibration error. The ECE is calculated and placed in the corner of the figure. FedWBA demonstrates the best calibration performance, ranking first in terms of ECE.

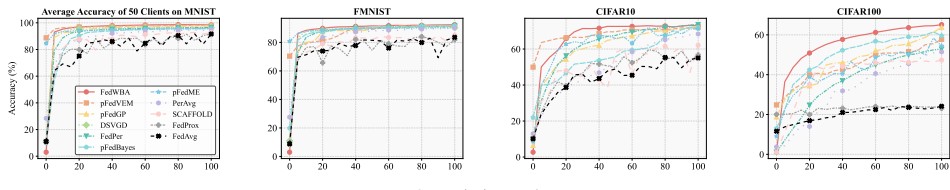

Figure 3: Comparison of convergence rates of different methods on MNIST, FMNIST, CIFAR-10, and CIFAR-100 with 50 clients. FedWBA exhibits the fastest convergence, with rapid growth in the first 10 communication rounds followed by steady improvement.

the number of clients increases and per client data volume is small, FedWBA can use its Bayesian features to learn and infer from limited data, resulting in better performance. Moreover, FedWBA has relatively small accuracy fluctuations. This stability is crucial in practical applications, especially in distributed learning with data heterogeneity and limited data, providing more reliable results.

## 6.3  Performance of Uncertainty Calibration

Benefiting from Bayesian principles, FedWBA provides superior predictive uncertainty quantification. Under identical experimental conditions as detailed in Section 6.1, we evaluate calibration performance through reliability diagrams in Figure 2. These diagrams compare model confidence (bars) against perfect calibration (diagonal), with FedWBA showing closest alignment. We also calculate the expected calibration error (ECE), which, as [16] suggests, measures the weighted average of empirical accuracy and model confidence. For the CIFAR-100 dataset, we present only the top-four performing methods; details of the remaining algorithms are provided in Appendix F.2. Results for additional datasets are also detailed in Appendix F.2, further corroborating the consistent superiority of our approach across diverse visual recognition tasks.

FedWBA attains a lower ECE by leveraging Bayesian principles that integrate prior knowledge and update beliefs based on observed data, which is essential for precise uncertainty quantification. Empowered by SVGD, it adapts well to data characteristics, enabling more accurate uncertainty quantification. In contrast, baseline models usually lack this adaptability, leading to subpar calibration and less accurate uncertainty quantification.

## 6.4  Convergence Rate

We compare the convergence rate of FedWBA with baseline models, updating local parameters 10 times per client before each server upload. The experiment involves 50 clients and runs for 100 communication rounds across MNIST, FMNIST, CIFAR-10, and CIFAR-100, with all other settings identical to those in Section 6.1. Test accuracy convergence curves for these datasets with 50 clients are shown in Figure 3, with additional details in Appendix F.3.

Clearly, FedWBA shows an excellent convergence rate. It improves rapidly in the first 10 rounds and then rises steadily until convergence. In contrast, some baseline models have slow convergence rates, and others like FedAvg have performance fluctuations due to data heterogeneity. The better convergence performance of FedWBA is attributed to the use of SVGD at the client side. SVGD enables our method to better adapt to local data, efficiently capture the local posterior distribution, and thereby improve the convergence efficiency in federated learning.

## 6.5 Ablation Study

We perform ablation studies to assess key components of the model under the setting of 100 clients, with all other configurations identical to those in Section 6.1, to deepen our understanding of the model's behavior. We consider (1) **Iteration number in SVGD**, (2) **Kernel selection in SVGD**, and (3) **Kernel bandwidth in SVGD**. Additionally, (4) **Number of labels per client**, (5) **Client scheduling ratio per communication round**, (6) **AdaGrad parameters for SVGD learning rate** and (7) **Kernel bandwidth in KDE** to estimate the global prior are also examined (Appendix F.4). Extended analysis in Appendix F.5 quantifies the accuracy-ECE-communication efficiency trade-offs, revealing practical implementation insights.

**Iteration number in SVGD**: Table 2 shows that accuracy improves with more SVGD iterations on MNIST and FMNIST but plateaus after about 30 and 50 iterations, respectively. This indicates that a large number of local iterations are unnecessary, avoiding excessive local computational pressure.

**Kernel selection in SVGD**: As shown in Table 2, we systematically evaluate Laplacian, sigmoid, and polynomial kernels under identical hyperparameters. Laplacian and RBF both employed the median heuristic for bandwidth selection, while the sigmoid kernel used scaling factor $\alpha = 1$ and bias $c = 0$, and the polynomial kernel adopted exponent $d = 2$. These results highlight the importance of kernel selection, with the RBF kernel yielding the best performance empirically.

**Kernel Bandwidth in SVGD**: As shown in Table 2, the Gaussian kernel bandwidth significantly affects SVGD performance. Small bandwidths yield high precision but cause unstable,

Table 2: Ablation studies were conducted to investigate the impact of the number of SVGD iterations, the choice of SVGD kernel, and kernel bandwidth on prediction performance. The experiments are performed using 100 clients on the MNIST and FMNIST datasets.

| MNIST | | FMNIST | |
|---|---|---|---|
| **Iteration Number in SVGD** | | | |
| Iteration Number | Acc(%) | Iteration Number | Acc(%) |
| 20 | 96.59 ± 0.01 | 40 | 91.49 ± 0.02 |
| 30 | 97.02 ± 0.03 | 50 | 91.62 ± 0.01 |
| 40 | **97.23 ± 0.01** | 60 | **91.65 ± 0.02** |
| **SVGD kernels** | | | |
| kernel | Acc(%) | kernel | Acc(%) |
| sigmoid | 93.90 ± 0.07 | sigmoid | 87.10 ± 0.02 |
| Laplacian | 97.12 ± 0.02 | Laplacian | 91.43 ± 0.02 |
| polynomial | 97.16 ± 0.01 | polynomial | 91.52 ± 0.05 |
| RBF | **97.23 ± 0.01** | RBF | **91.65 ± 0.02** |
| **Kernel Bandwidth in SVGD** | | | |
| bandwidth | Acc(%) | bandwidth | Acc(%) |
| 1 | **97.34 ± 0.05** | 1 | **91.71 ± 0.10** |
| med | 97.26 ± 0.02 | med | 91.65 ± 0.02 |
| 12 | 97.10 ± 0.03 | 12 | 91.22 ± 0.03 |

fluctuating approximations and oversensitivity to local noise. Large bandwidths favor global exploration yet obscure local details, slowing convergence. Using the median heuristic effectively balances these aspects, supporting both exploration and local refinement for stable, efficient convergence.

The findings guide configuring FedWBA's Bayesian components, showing deliberate selection of SVGD iterations, kernel type, and bandwidth is critical for optimal accuracy in FL.

## 7 Conclusions

In summary, we propose FedWBA, a novel PBFL method that simultaneously enhances local inference and global aggregation. At the client side, we employ particle-based variational inference for nonparametric posterior representation. This allows clients to capture complex local posterior distributions by leveraging the flexibility of particle-based methods, better approximating the posterior of model parameters. At the server side, we introduce particle-based Wasserstein barycenter aggregation, offering a more geometrically meaningful way to aggregate local client updates. Theoretically, we establish convergence guarantees for FedWBA. Comprehensive experiments reveal that FedWBA outperforms baselines in prediction accuracy, uncertainty calibration, and convergence rate.

## Acknowledgments

This work was supported by the NSFC Projects (No.12171479, No.62576346, No.12371516), the MOE Project of Key Research Institute of Humanities and Social Sciences (22JJD110001), the fundamental research funds for the central universities, and the research funds of Renmin University of China (24XNKJ13), and Beijing Advanced Innovation Center for Future Blockchain and Privacy Computing.

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

## A  Comparison of Aggregation Methods

Figure 4 presents three different aggregation methods for three Gaussian distributions: parameter averaging, mixture, and Wasserstein barycenter. Notably, both parameter averaging and the Wasserstein barycenter result in Gaussian distributions, while the mixture does not. The choice of Gaussian distributions is motivated by the Gaussian assumption imposed by pFedBayes on the variational parameter family, which serves as a representative baseline in our comparative analysis of Bayesian federated learning approaches.

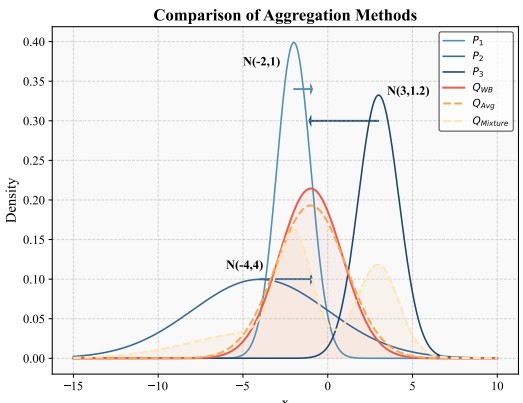

Figure 4: Comparison of Three Aggregation Methods: Wasserstein Barycenter (WB), Parameter Averaging (Avg), and Arithmetic Mean (Mixture).

## B  Proof of Equation (5)

Define $\boldsymbol{\Theta} \overset{\text{def}}{=} \text{diag}(\boldsymbol{\Theta}^\top \boldsymbol{\Theta})$ and $\boldsymbol{\Theta}_k \overset{\text{def}}{=} \text{diag}(\boldsymbol{\Theta}_k^\top \boldsymbol{\Theta}_k)$. Then, let $\mathbf{M}_k$ denote the squared Euclidean distance matrix between the pairwise elements of $\boldsymbol{\Theta}$ and $\boldsymbol{\Theta}_k$, which can be written as:

$$\mathbf{M}_k = \boldsymbol{\Theta}\mathbf{1}_N^\top + \mathbf{1}_N\boldsymbol{\Theta}_k^\top - 2\boldsymbol{\Theta}\boldsymbol{\Theta}_k^\top. \tag{8}$$

Following this form of $\mathbf{M}_k$, the Frobenius inner product between $\mathbf{M}_k$ and $\mathbf{T}_k$ is given by:

$$\langle \mathbf{M}_k, \mathbf{T}_k \rangle_F = \text{tr}(\mathbf{T}_k^\top \boldsymbol{\Theta}\mathbf{1}_N^\top) + \text{tr}(\mathbf{T}_k^\top \mathbf{1}_N\boldsymbol{\Theta}_k^\top) - 2\text{tr}(\mathbf{T}_k^\top \boldsymbol{\Theta}\boldsymbol{\Theta}_k^\top) \tag{9}$$

$$= \frac{1}{N}\text{tr}(\boldsymbol{\Theta}) + \frac{1}{N}\text{tr}(\boldsymbol{\Theta}_k^\top) - 2\text{tr}(\mathbf{T}_k^\top \boldsymbol{\Theta}\boldsymbol{\Theta}_k^\top). \tag{10}$$

By minimizing Equation (9), and taking into account that $\boldsymbol{\Theta}_k$ is a constant, we obtain:

$$\mathbf{T}_k^* = \underset{\mathbf{T}_k \in \mathcal{T}}{\arg\min} \langle \mathbf{M}_k, \mathbf{T}_k \rangle_F = \underset{\mathbf{T}_k \in \mathcal{T}}{\arg\min} \frac{1}{N}\text{tr}(\boldsymbol{\Theta}) - 2\langle \boldsymbol{\Theta}, \mathbf{T}_k\boldsymbol{\Theta}_k \rangle. \tag{11}$$

After obtain $\mathbf{T}_k^*$, we can obtain the global prior as the Wasserstein barycenter of $K$ local posteriors:

$$\hat{\hat{p}}^*(\boldsymbol{\theta}) = \underset{\boldsymbol{\Theta}}{\arg\min} \frac{1}{K} \sum_{k=1}^{K} \langle \mathbf{M}_k, \mathbf{T}_k^* \rangle_F$$

$$= \underset{\boldsymbol{\Theta}}{\arg\min} \frac{1}{K} \sum_{k=1}^{K} \left\{ \|\boldsymbol{\Theta}\text{diag}(N^{1/2}) - \mathbf{T}_k^*\boldsymbol{\Theta}_k\text{diag}(N^{-1/2})\|^2 - \|\mathbf{T}_k^*\boldsymbol{\Theta}_k\text{diag}(N^{-1/2})\|^2 \right\}. \tag{12}$$

Consequently, the optimal $\boldsymbol{\Theta}^*$ has an analytical solution

$$\boldsymbol{\Theta}^* = \frac{1}{K} \sum_{k=1}^{K} \mathbf{T}_k^*\boldsymbol{\Theta}_k\text{diag}(N^{-1}). \tag{13}$$

$\square$

## C   Pseudocode for Algorithm

---
**Algorithm 1** FedWBA
---

**Input:** Size $Z$, kernels $k(\cdot, \cdot)$ and $\tilde{k}(\cdot, \cdot)$
**Initialize:** local posterior particles $\{\{\boldsymbol{\theta}_{k,i}\}_{i=1}^N\}_{k=1}^K$; global prior particles $\{\boldsymbol{\theta}_i\}_{i=1}^N$
   **for** $\langle$communication round$\rangle$
      $\mathcal{K} \leftarrow$ Server randomly samples a subset of clients of size $Z$;
      Server broadcasts $\{\boldsymbol{\theta}_i\}_{i=1}^N$ to each client $k \in \mathcal{K}$;
**[Local updates]**
      **for** client $k$ **in** $\mathcal{K}$ **do**
         Rebuild the continuous prior by Equation (6);
         Update local posterior particles by Equation (2);
      **end for**
      Each client $k \in \mathcal{K}$ upload $\{\boldsymbol{\theta}_{k,i}\}_{i=1}^N$ to the server;
**[Server aggregates]**
      Update global prior particles by Equation (5).

---

## D   Proof of Theorem 5.3

*Proof.* The ELBO $F(q_k(\boldsymbol{\theta}))$ can be expressed as the negative KL divergence: $F(q_k(\boldsymbol{\theta})) = -\text{KL}\big(q_k(\boldsymbol{\theta}) \parallel \tilde{p}(\boldsymbol{\theta} \mid D_k)\big)$, where $\tilde{p}(\boldsymbol{\theta} \mid D_k) = p(\mathcal{D}_k|\boldsymbol{\theta})\overline{p}(\boldsymbol{\theta})$ is the unnormalized posterior distribution.

For iteration $l$ on client $k$, the change in ELBO from iteration $l$ to $l+1$ is given by:

$$
\begin{aligned}
F(q_k^{(l+1)}(\boldsymbol{\theta})) - F(q_k^{(l)}(\boldsymbol{\theta})) &= \text{KL}\big(q_k^{(l)}(\boldsymbol{\theta})\|\tilde{p}(\boldsymbol{\theta} \mid \mathcal{D}_k)\big) - \text{KL}\big(q_k^{(l+1)}(\boldsymbol{\theta})\|\tilde{p}(\boldsymbol{\theta} \mid \mathcal{D}_k)\big) \\
&\overset{\text{(i)}}{=} \text{KL}\Big(q_k^{(l)}(\boldsymbol{\theta})\|\tilde{p}(\boldsymbol{\theta} \mid \mathcal{D}_k)\Big) - \text{KL}\Big(q_k^{(l)}(\boldsymbol{\theta})\|\mathbf{t}^{-1}\big(\tilde{p}(\boldsymbol{\theta} \mid \mathcal{D}_k)\big)\Big) \\
&= \mathbb{E}_{q_k^{(l)}(\boldsymbol{\theta})}\Big[ -\log \tilde{p}(\boldsymbol{\theta} \mid \mathcal{D}_k) + \log \mathbf{t}^{-1}\big(\tilde{p}(\boldsymbol{\theta} \mid \mathcal{D}_k)\big)\Big] \\
&\overset{\text{(ii)}}{=} \mathbb{E}_{q_k^{(l)}(\boldsymbol{\theta})}\Big[ -\log \tilde{p}(\boldsymbol{\theta} \mid \mathcal{D}_k) \\
&\qquad + \log \tilde{p}\big(\mathbf{t}(\boldsymbol{\theta}) \mid \mathcal{D}_k\big) + \log \Big| \det \big(\nabla_{\boldsymbol{\theta}}\mathbf{t}(\boldsymbol{\theta})\big)\Big|\Big].
\end{aligned}
\tag{14}
$$

(i) is obtained from the SVGD based particle transformation $\boldsymbol{\theta}^{(l+1)} = \mathbf{t}(\boldsymbol{\theta}^{(l)}) = \boldsymbol{\theta}^{(l)} + \epsilon\phi(\boldsymbol{\theta}^{(l)})$ and the definition of KL divergence. (ii) follows from the change of variable formula for densities $\mathbf{t}^{-1}(p(\boldsymbol{\theta})) = p(\mathbf{t}(\boldsymbol{\theta}))|\det\big(\nabla_{\boldsymbol{\theta}}\mathbf{t}(\boldsymbol{\theta})\big)|$.

Applying a first-order Taylor expansion for $\boldsymbol{\theta}$:

$$
\log \tilde{p}\big(\mathbf{t}(\boldsymbol{\theta}) \mid \mathcal{D}_k\big) \approx \log \tilde{p}(\boldsymbol{\theta} \mid \mathcal{D}_k) + \epsilon\nabla_{\boldsymbol{\theta}}\log \tilde{p}(\boldsymbol{\theta} \mid \mathcal{D}_k)^\top\phi_k(\boldsymbol{\theta}) + \frac{1}{2}\epsilon^2\nabla_{\boldsymbol{\theta}}^2\log \tilde{p}(\boldsymbol{\theta} \mid \mathcal{D}_k)\phi_k^2(\boldsymbol{\theta}). \tag{15}
$$

According to Assumption 5.1, we have

$$
\log|\det\big(\nabla_{\boldsymbol{\theta}}\mathbf{t}(\boldsymbol{\theta})\big)| = \sum_{i=1}^D \log|e_i| \geq \sum_{i=1}^D (1 - e_i^{-1}) = \text{tr}\Big(\boldsymbol{I} - \big(\nabla_{\boldsymbol{\theta}}\mathbf{t}(\boldsymbol{\theta})\big)^{-1}\Big). \tag{16}
$$

Given that $\nabla_{\boldsymbol{\theta}}\mathbf{t}(\boldsymbol{\theta}) = \boldsymbol{I} + \epsilon\nabla_{\boldsymbol{\theta}}\phi(\boldsymbol{\theta})$, and under Assumption 5.2, applying the Neumann expansion, we get the approximation of $\big(\nabla_{\boldsymbol{\theta}}\mathbf{t}(\boldsymbol{\theta})\big)^{-1}$:

$$
\big(\nabla_{\boldsymbol{\theta}}\mathbf{t}(\boldsymbol{\theta})\big)^{-1} \approx \boldsymbol{I} - \epsilon\nabla_{\boldsymbol{\theta}}\phi(\boldsymbol{\theta}) + \big(\epsilon\nabla_{\boldsymbol{\theta}}\phi(\boldsymbol{\theta})\big)^2. \tag{17}
$$

Then Equation (16) simplifies to:

$$
\log|\det\big(\nabla_{\boldsymbol{\theta}}\mathbf{t}(\boldsymbol{\theta})\big)| \geq \text{tr}\Big(\epsilon\nabla_{\boldsymbol{\theta}}\phi(\boldsymbol{\theta}) - \big(\epsilon\nabla_{\boldsymbol{\theta}}\phi(\boldsymbol{\theta})\big)^2\Big). \tag{18}
$$

Substituting Equations (15) and (18) into Equation (14), we obtain:

$$
\begin{aligned}
F(q_k^{(l+1)}(\boldsymbol{\theta})) - F(q_k^{(l)}(\boldsymbol{\theta})) &\geq \epsilon \mathbb{E}_{q_k^{(l)}(\boldsymbol{\theta})} \left[ \mathrm{tr}\Big( \nabla_{\boldsymbol{\theta}} \log \tilde{p}(\boldsymbol{\theta} \mid \mathcal{D}_k)^\top \phi_k(\boldsymbol{\theta}) + \nabla_{\boldsymbol{\theta}} \phi_k(\boldsymbol{\theta}) \Big) \right] \\
&\quad - \epsilon^2 \mathbb{E}_{q_k^{(l)}(\boldsymbol{\theta})} \left[ \mathrm{tr}\Big( \big( \nabla_{\boldsymbol{\theta}} \phi_k(\boldsymbol{\theta}) \big)^2 - \frac{1}{2} \nabla_{\boldsymbol{\theta}}^2 \log \tilde{p}(\boldsymbol{\theta} \mid \mathcal{D}_k) \phi_k^2(\boldsymbol{\theta}) \Big) \right] \\
&\geq \epsilon D\big( q_k^{(l)}(\boldsymbol{\theta}), \tilde{p}(\boldsymbol{\theta} \mid D_k) \big) \\
&\quad - \epsilon^2 \mathbb{E}_{q_k^{(l)}(\boldsymbol{\theta})} \left[ \mathrm{tr}\Big( \big( \nabla_{\boldsymbol{\theta}} \phi_k(\boldsymbol{\theta}) \big)^2 - \frac{1}{2} \nabla_{\boldsymbol{\theta}}^2 \log \tilde{p}(\boldsymbol{\theta} \mid \mathcal{D}_k) \phi_k^2(\boldsymbol{\theta}) \Big) \right].
\end{aligned}
\tag{19}
$$

Given that the learning rate $\epsilon$ is sufficiently small, the second term becomes negligible. Therefore, the lower bound of the ELBO increase per iteration is:

$$
F(q_k^{(l+1)}(\boldsymbol{\theta})) - F(q_k^{(l)}(\boldsymbol{\theta})) \geq \epsilon D\big( q_k^{(l)}(\boldsymbol{\theta}), \tilde{p}(\boldsymbol{\theta} \mid \mathcal{D}_k) \big).
\tag{20}
$$

$\square$

# E  Proof of Theorem 5.9

## E.1  Preliminaries

We begin by recalling the definition of Pseudo Hellinger distance and then state key lemmas that are essential for the proof of Theorem 5.9.

**Definition E.1. (Pseudo Hellinger distance)** The pseudo Hellinger distance between probability measures $P_{\boldsymbol{\theta}}, P_{\boldsymbol{\theta}'}$ is

$$
h_{sk}^2(\boldsymbol{\theta}, \boldsymbol{\theta}') = \frac{1}{s} \sum_{i=1}^s h^2 \left\{ p(D_{ki} \mid \boldsymbol{\theta}), p(D_{ki} \mid \boldsymbol{\theta}') \right\},
\tag{21}
$$

where $i$ represents the data index and $h(p_1, p_2) = \left[ \int \left\{ \sqrt{p_1(y)} - \sqrt{p_2(y)} \right\}^2 \mathrm{d}y \right]^{1/2}$ is the Hellinger distance between two generic densities $p_1, p_2$.

**Lemma E.2.** *(Generalization of [41] Theorem 1) Assume Assumption 5.6 holds. Then for any $\delta > 0$, there exist positive constants $q_1$, $q_2$ that depend on $C_1$, $C_2$, such that for all subsets $D_k$ with $k = 1, \ldots, K$ and all sufficiently large $s$,*

$$
P_{\boldsymbol{\theta}_0}^{(S)} \left( \sup_{h_{sk}(\boldsymbol{\theta}, \boldsymbol{\theta}_0) \geq \delta} \prod_{i=1}^s \frac{p(D_{ki}|\boldsymbol{\theta})}{p(D_{ki}|\boldsymbol{\theta}_0)} \geq \exp(-q_1 s \delta^2) \right) \leq 4 \exp(-q_2 s \delta^2).
$$

**Lemma E.3.** *Assume Assumption 5.6 holds. Then for any $\delta > 0$, there exist positive constants $r_1$, $r_2$ that depend on $\kappa$, $c_\pi$, such that for every subset $D_k$ ($k = 1, \ldots, K$), for any $t \geq \eta_s^{2\alpha}$,*

$$
P_{\boldsymbol{\theta}_0}^{(S)} \left( \int_\Xi \prod_{i=1}^s \frac{p(D_{ki}|\boldsymbol{\theta})}{p(D_{ki}|\boldsymbol{\theta}_0)} \Pi(\mathrm{d}\boldsymbol{\theta}) \leq \exp(-r_1 St) \right) \leq \exp(-r_2 st).
$$

**Lemma E.4.** *Let $\overline{\boldsymbol{\nu}}$ denote the $W_2$ barycenter of $N$ measures $\boldsymbol{\nu}_1, \ldots, \boldsymbol{\nu}_N$ in $\mathcal{P}_2(\Xi)$. Then for any $\boldsymbol{\theta}_0 \in \Xi$, the following inequality holds:*

$$
W_2(\overline{\boldsymbol{\nu}}, \delta_{\boldsymbol{\theta}_0}) \leq \frac{1}{N} \sum_{j=1}^N W_2(\boldsymbol{\nu}_j, \delta_{\boldsymbol{\theta}_0}).
$$

This lemma establishes a relationship between the barycenter and the distances from each individual measure to a fixed point $\boldsymbol{\theta}_0$. The proofs of Lemmas E.2 to E.4 can be found in [36].

**Lemma E.5.** *Suppose that Assumption 5.4-Assumption 5.8 hold for the $k$-th subset posterior $p(\boldsymbol{\theta}|D_k)$ with $k = 1, \ldots, K$. Then there exists a constant $C_3$ that depends on $C_L$, $C_1$, $C_2$, $\kappa$, $c_\pi$ and does not depend on $k$, such that as $s \to \infty$,*

$$
\mathbb{E}_{P_{\boldsymbol{\theta}_0}} W_2^2 \Big( p(\boldsymbol{\theta}|D_k), \delta_{\boldsymbol{\theta}_0}(\cdot) \Big) \leq C_3 \left( \frac{\log^2 s}{s} \right)^{\frac{1}{\alpha}}.
$$

*Proof.* Let $\eta_s = (\frac{s}{\log^2 s})^{-\frac{1}{2\alpha}}$. Due to the compactness of $\Xi$ in assumption Assumption 5.4, there exists a large finite constant $M_0$ such that $\rho(\boldsymbol{\theta}, \boldsymbol{\theta}_0) \leq M_0$. We begin with a decomposition of the $W_2$ distance from the $k$-th subset posterior $p(\boldsymbol{\theta}|D_k)$ to the delta measure at the true parameter $\boldsymbol{\theta}_0$:

$$
\begin{aligned}
\mathbb{E}_{P_{\boldsymbol{\theta}_0}} W_2^2\big(p(\boldsymbol{\theta}|D_k), \delta_{\boldsymbol{\theta}_0}(\cdot)\big) &= \mathbb{E}_{P_{\boldsymbol{\theta}_0}} \int_\Xi \rho^2(\boldsymbol{\theta}, \boldsymbol{\theta}_0) p(\mathrm{d}\boldsymbol{\theta}|D_k) \\
&\leq \mathbb{E}_{P_{\boldsymbol{\theta}_0}} \int_{\{\boldsymbol{\theta}: \rho(\boldsymbol{\theta}, \boldsymbol{\theta}_0) \leq C_4 \eta_s\}} \rho^2(\boldsymbol{\theta}, \boldsymbol{\theta}_0) p(\mathrm{d}\boldsymbol{\theta}|D_k) + \mathbb{E}_{P_{\boldsymbol{\theta}_0}} \int_{\{\boldsymbol{\theta}: \rho(\boldsymbol{\theta}, \boldsymbol{\theta}_0) > C_4 \eta_s\}} \rho^2(\boldsymbol{\theta}, \boldsymbol{\theta}_0) p(\mathrm{d}\boldsymbol{\theta}|D_k) \\
&\leq (C_4 \eta_s)^2 + M_0^2 \mathbb{E}_{P_{\boldsymbol{\theta}_0}} p\big(\rho(\boldsymbol{\theta}, \boldsymbol{\theta}_0) > C_4 \eta_s | D_k\big).
\end{aligned}
\tag{22}
$$

We will choose the constant $C_4$ as $C_4 = (\frac{2 r_1 K}{q_1 C_L})^{\frac{1}{2\alpha}}$, where $C_L$, $q_1$, $r_1$ are the constants in Assumption 5.4, Assumption 5.5, Lemma E.2 and Lemma E.3. Using Assumption 5.5, we can further replace the $\rho$ metric by the pseudo Hellinger distance:

$$
\begin{aligned}
p\big(\theta \in \Xi : \rho(\boldsymbol{\theta}, \boldsymbol{\theta}_0) > C_4 \eta_s | D_k\big) &\leq p\big(\theta \in \Xi : h_{sk}(P_{\boldsymbol{\theta},k}, P_{\boldsymbol{\theta}_0,k}) > \sqrt{C_L}(C_4 \eta_s)^\alpha | D_k\big) \\
&= \int_{\{\boldsymbol{\theta} \in \Xi : h_{sk}(\boldsymbol{\theta}, \boldsymbol{\theta}_0) > \sqrt{\frac{2 r_1 K}{q_1}} \eta_s^\alpha\}} \frac{\prod_{i=1}^s \left[\frac{p(D_{ki}|\boldsymbol{\theta})}{p(D_{ki}|\boldsymbol{\theta}_0)}\right] \Pi(\mathrm{d}\boldsymbol{\theta})}{\int_\Xi \prod_{i=1}^s \left[\frac{p(D_{ki}|\boldsymbol{\theta})}{p(D_{ki}|\boldsymbol{\theta}_0)}\right] \Pi(\mathrm{d}\boldsymbol{\theta})}.
\end{aligned}
\tag{23}
$$

For the denominator in Equation (22), by Assumption 5.7 and Lemma E.3, when $s$ is sufficiently large, with probability at least $1 - \exp(-r_2 s \eta_s^{2\alpha})$

$$
\int_\Xi \prod_{i=1}^s \frac{p(D_{ki}|\boldsymbol{\theta})}{p(D_{ki}|\boldsymbol{\theta}_0)} \Pi(\mathrm{d}\boldsymbol{\theta}) > \exp(-r_1 S \eta_s^{2\alpha}).
\tag{24}
$$

For the numerator in Equation (23), by Assumption 5.6 and Lemma E.2, setting $\delta = \sqrt{\frac{2 r_1 K}{q_1}} \eta_s^\alpha$, we get that with probability at least $1 - 4 \exp\left(-\frac{2 r_1 q_2}{q_1} S \eta_s^{2\alpha}\right)$,

$$
\sup_{\{\boldsymbol{\theta} \in \Xi : h_{sk}(\boldsymbol{\theta}, \boldsymbol{\theta}_0) \geq \sqrt{\frac{2 r_1 K}{q_1}} \eta_s^\alpha\}} \prod_{i=1}^s \left[\frac{p(D_{ki}|\boldsymbol{\theta})}{p(D_{ki}|\boldsymbol{\theta}_0)}\right] \leq \exp(-2 r_1 S \eta_s^{2\alpha}).
\tag{25}
$$

Therefore, based on Equations (23) to (25), with probability at least $1 - 4 \exp\left(-\frac{2 r_1 q_2}{q_1} S \eta_s^{2\alpha}\right) - \exp(-r_2 s \eta_s^{2\alpha})$,

$$
p(\boldsymbol{\theta} \in \Xi : \rho(\boldsymbol{\theta}, \boldsymbol{\theta}_0) > C_4 \eta_s \mid D_k) \leq \exp(-2 r_1 S \eta_s^{2\alpha} + r_1 S \eta_s^{2\alpha}) \leq \exp(-r_1 S \eta_s^{2\alpha}).
\tag{26}
$$

Let $A_{\eta_s}$ be the event $\left\{\boldsymbol{\theta} \in \Xi : p(\boldsymbol{\theta} \in \Xi : \rho(\boldsymbol{\theta}, \boldsymbol{\theta}_0) > C_4 \eta_s \mid D_k) \leq \exp(-r_1 S \eta_s^{2\alpha})\right\}$. Then we can bound the second term in Equation (22) as follows:

$$
\begin{aligned}
\mathbb{E}_{P_{\boldsymbol{\theta}_0}} p\big(\rho(\boldsymbol{\theta}, \boldsymbol{\theta}_0) > C_4 \eta_s \mid D_k\big) &\leq \mathbb{E}_{P_{\boldsymbol{\theta}_0}}\left[I(A_{\eta_s}) p\big(\rho(\boldsymbol{\theta}, \boldsymbol{\theta}_0) > C_4 \eta_s \mid D_k\big)\right] \\
&\quad + \mathbb{E}_{P_{\boldsymbol{\theta}_0}}\left[I(A_{\eta_s}^c) p\big(\rho(\boldsymbol{\theta}, \boldsymbol{\theta}_0) > C_4 \eta_s \mid D_k\big)\right] \\
&\leq \exp(-r_1 S \eta_s^{2\alpha}) + 4 \exp\left(-\frac{2 r_1 q_2}{q_1} S \eta_s^{2\alpha}\right) \\
&\quad + \exp(-r_2 s \eta_s^{2\alpha}) \\
&\leq 6 \exp(-C_5 s \eta_s^{2\alpha}),
\end{aligned}
\tag{27}
$$

for $C_5 = \min(r_1, r_2, \frac{2 r_1 q_2}{q_1})$, as clearly the second term is dominating the other two given $m \lesssim n$.

Therefore, for Equation (22), since $\eta_s = (s/\log_2 s)^{-\frac{1}{2\alpha}}$, as $s \to \infty$, an explicit bound will be

$$\mathbb{E}_{P_{\boldsymbol{\theta}_0}} W_2^2\Big(p(\boldsymbol{\theta}|D_k), \delta_{\boldsymbol{\theta}_0}(\cdot)\Big) \leq C_4^2 \frac{\log^{\frac{2}{\alpha}} s}{s^{\frac{1}{\alpha}}} + 6M_0^2 \exp(-c_2 \log^2 s)$$

$$\leq C_4^2 \frac{\log^{\frac{2}{\alpha}} s}{s^{\frac{1}{\alpha}}} + \frac{1}{s^{1+\frac{1}{\alpha}}} \tag{28}$$

$$\leq C \frac{\log^{\frac{2}{\alpha}} s}{s^{\frac{1}{\alpha}}},$$

as $s$ becomes sufficiently large, where the constant $C$ depends on $\alpha$, $C_4$, $C_5$, which further depends on $q_1$, $q_2$, $r_1$, $r_2$, $C_L$. Since $q_1$, $q_2$ in Lemma E.2 and $r_1$, $r_2$ in Lemma E.3 depend on $C_1$, $C_2$, $\kappa$, $c_\pi$, it follows that $C_3$ depends on $C_L$, $C_1$, $C_2$, $\kappa$, $c_\pi$. □

## E.2 Main Proof

Now we give the full proof of Theorem 5.9.

*Proof.* For notational simplicity in the subsequent derivations, we abbreviate the Wasserstein barycenter distribution $\overline{p}(\boldsymbol{\theta}|D_k)$ as $\overline{p}(\boldsymbol{\theta})$. Following Lemma E.5, we have:

$$P_{\boldsymbol{\theta}_0}^{(S)}\left(W_2\big(\overline{p}(\boldsymbol{\theta}), \delta_{\theta_0}(\cdot)\big) > \sqrt{C \frac{\log^{\frac{2}{\alpha}} s}{s^{\frac{1}{\alpha}}}}\right)$$

$$\leq P_{\boldsymbol{\theta}_0}^{(S)}\left(\frac{1}{K}\sum_{k=1}^{K} W_2\big(p(\boldsymbol{\theta}|D_k), \delta_{\theta_0}(\cdot)\big) > \sqrt{C \frac{\log^{\frac{2}{\alpha}} s}{s^{\frac{1}{\alpha}}}}\right). \tag{29}$$

Next, applying Markov's inequality and using the relation between $l_1$ and $l_2$ norms, we obtain:

$$P_{\boldsymbol{\theta}_0}^{(S)}\left(W_2(p(\boldsymbol{\theta}|D_k), \delta_{\theta_0}(\cdot)) > \sqrt{C \frac{\log^{\frac{2}{\alpha}} s}{s^{\frac{1}{\alpha}}}}\right) \leq \frac{1}{\frac{C\log^{2/\alpha} s}{s^{1/\alpha}}} \mathbb{E}_{P_{\boldsymbol{\theta}_0}}\left[\frac{1}{K}\sum_{k=1}^{K} W_2\big(p(\boldsymbol{\theta}|D_k), \delta_{\theta_0}(\cdot)\big)\right]^2$$

$$\tag{30}$$

$$\leq \frac{s^{1/\alpha}}{CK\log^{2/\alpha} s}\sum_{k=1}^{K} \mathbb{E}_{P_{\boldsymbol{\theta}_0}} W_2^2\big(p(\boldsymbol{\theta}|D_k), \delta_{\theta_0}(\cdot)\big) \tag{31}$$

$$\leq \frac{s^{\frac{1}{\alpha}}}{CK\log^{\frac{2}{\alpha}} s} \cdot KC_1 \frac{\log^{\frac{2}{\alpha}} s}{s^{\frac{1}{\alpha}}} \tag{32}$$

$$= \frac{C_1}{C}, \tag{33}$$

where the second step inequality follows from the fact that for nonnegative real-valued random variables $X_1, \cdots, X_n$, $(\frac{1}{n}\sum_{i=1}^n X_i)^2 \leq \frac{1}{n}\sum_{i=1}^n X_i^2$, and the third step inequality uses the previously obtained upper bound $\mathbb{E}_{P_{\boldsymbol{\theta}_0}} W_2^2\big(p(\boldsymbol{\theta}|D_k), \delta_{\theta_0}(\cdot)\big) \leq C_1 \frac{\log^{\frac{2}{\alpha}} s}{s^{\frac{1}{\alpha}}}$.

Finally, combing these results, we conclude:

$$W_2\Big(\overline{p}(\boldsymbol{\theta}), \delta_{\boldsymbol{\theta}_0}(\cdot)\Big) = O_p\Big(\sqrt{\frac{\log^{\frac{2}{\alpha}} s}{s^{\frac{1}{\alpha}}}}\Big). \tag{34}$$

□

## F More Experimental Results

In this section, we provide more experimental results to demonstrate the performance of our method compared to others.

### F.1 Performance of Global Model

To validate the performance of the global model, we compare with FedAvg, FedProx, and SCAFFOLD on MNIST, using settings identical to Section 6.1. The results show a highly comparable performance, demonstrating the validity of the global model as a prior regularizer.

Table 3: Global model performance with best results bolded.

| DATASET | METHOD | 50 CLIENTS | 100 CLIENTS | 200 CLIENTS |
|---------|--------|------------|-------------|-------------|
| MNIST | FEDAVG | $91.98 \pm 0.07$ | $91.76 \pm 0.08$ | $\mathbf{90.94 \pm 0.06}$ |
| | FEDPROX | $92.12 \pm 0.08$ | $92.04 \pm 0.11$ | $90.82 \pm 0.16$ |
| | SCAFFOLD | $\mathbf{92.90 \pm 0.07}$ | $\mathbf{92.14 \pm 0.08}$ | $90.85 \pm 0.11$ |
| | OURS | $92.02 \pm 0.02$ | $91.95 \pm 0.03$ | $89.90 \pm 0.03$ |

### F.2 Performance of Uncertainty Quantification

In this section, we present the performance of the remaining six algorithms in uncertainty quantification, as shown in Figure 5.

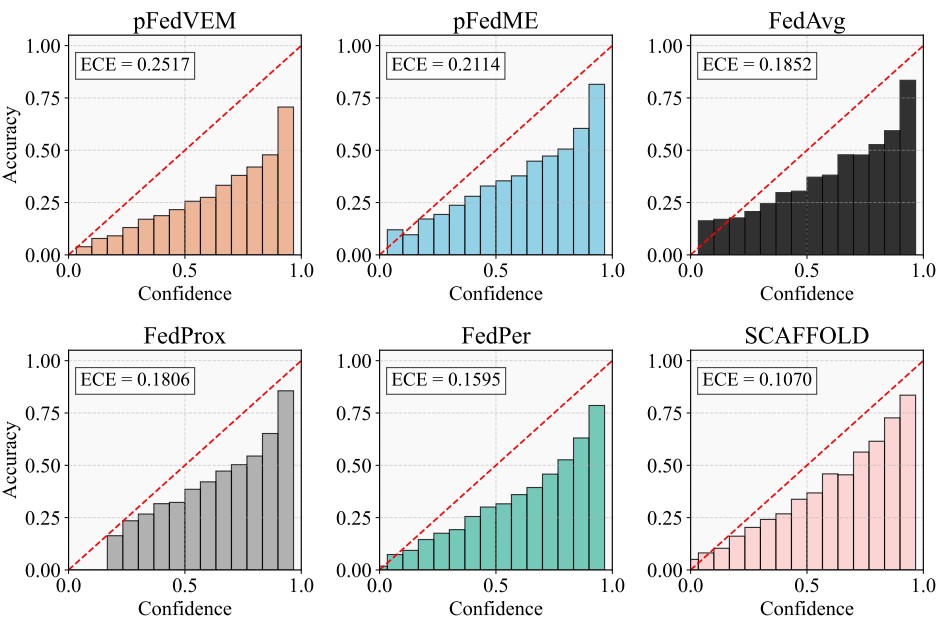

Figure 5: Reliability diagrams of the six methods on CIFAR-100. The perfect calibration is plotted as a red diagonal line, and the actual results are presented as bar charts. The gap between the top of each bar and the red line represents the calibration error. The ECE is calculated and placed in the top-left corner of the figure. Among them, the method with the highest ECE value has the worst calibration performance.

Our uncertainty quantification experiments across MNIST, FMNIST, and CIFAR-10 demonstrate that our method achieves state-of-the-art performance in ECE, outperforming existing baselines on all benchmarks, as presented in Table 4. This consistent superiority highlights enhanced generalization and calibration capabilities under varying data distributions and task complexities.

### F.3 Convergence Rate

This section presents a comparison of algorithm convergence across MNIST, FMNIST, CIFAR-10, and CIFAR-100 with 100 and 200 clients (as depicted in Figure 6 for 100 clients and Figure 7 for 200 clients). Similar to observations in prior client configurations, FedWBA exhibits rapid test accuracy

Table 4: ECE of different methods on MNIST, FMNIST and CIFAR-10 with 50 clients. Optimal results are  **bolded** .

| method | Dataset | | |
|--------|---------|---------|----------|
| | MNIST | FMNIST | CIFAR-10 |
| FedAvg | 0.0449 | 0.0211 | 0.0394 |
| FedProx | 0.0147 | 0.0308 | 0.0264 |
| SCAFFOLD | 0.0424 | 0.0343 | 0.1002 |
| FedPer | 0.0061 | 0.0082 | 0.0284 |
| perFedAvg | 0.0213 | 0.0258 | 0.0663 |
| pFedME | 0.0097 | 0.0424 | 0.0549 |
| pFedBayes | 0.0158 | 0.0132 | 0.0420 |
| pFedVEM | 0.0130 | 0.0107 | 0.0476 |
| pFedGP | 0.0217 | 0.0193 | 0.0245 |
| Ours | **0.0014** | **0.0078** | **0.0132** |

growth within the first 10 communication rounds across all datasets, followed by gradual refinement and eventual convergence. This consistent pattern across diverse datasets and client scales highlights the approach's effectiveness and stability in federated learning, demonstrating robust adaptability to varying data distribution complexities.

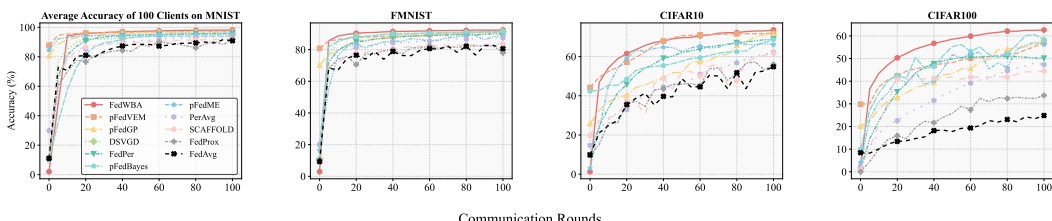

Figure 6: Comparison of convergence rates of different methods on MNIST, FMNIST, CIFAR-10, and CIFAR-100 with 100 clients.

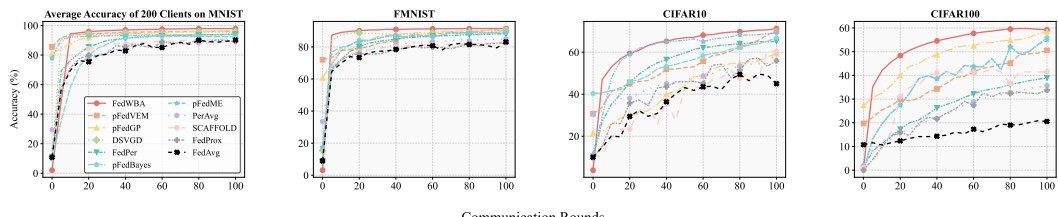

Figure 7: Comparison of convergence rates of different methods on MNIST, FMNIST, CIFAR-10, and CIFAR-100 with 200 clients.

### F.4 Ablation Study

In this section, we conduct ablation studies on four key components: (1) **Number of labels per client**, (2) **Client scheduling ratio per communication round**, (3) **two AdaGrad parameters for SVGD learning rate**, and (4) **Kernel bandwidth in KDE** to estimate the global prior. In the optimization algorithm, the AdaGrad update rule is:

$$\boldsymbol{\theta}^{(l+1)} = \boldsymbol{\theta}^{(l)} - \frac{\eta}{\sqrt{\boldsymbol{G}^l + \lambda}} \odot \boldsymbol{g}^l,$$

where $\boldsymbol{G}^l$ is the accumulated sum of squared gradients up to the $l$-th iteration, $\boldsymbol{g}^l$ is the gradient vector at the $l$-th iteration, $\eta$ is the global learning rate determining step-size, and $\lambda$ is a smoothing

term that prevents the denominator from being zero and enhances algorithm stability, which is crucial for SVGD learning rate adjustment.

**Number of labels per client**: To validate the method's generality across varying client problem complexities, we evaluate MNIST and FMNIST with 2, 5, and 10 labels per client—specifically, we construct client heterogeneity in terms of label distribution through this varying number of labels per client, simulating real-world scenarios where clients only hold partial and unequal label information. Smaller label counts indicate simpler client tasks. As shown in Table 5, accuracy decreases with increasing labels, aligning with expected difficulty trends. Despite performance variation with task complexity, FedWBA maintains stable convergence and high accuracy at each label setting, demonstrating robust adaptability to diverse client-side problem scales.

**Client scheduling ratio per communication round**: We conduct experiments on MNIST and FMNIST with client scheduling ratios of 0.1, 0.2, and 0.5 per communication round. Results show that a higher scheduling ratio correlates with faster convergence of the global prior and higher client-side accuracy—this is attributed to more client updates contributing to the global model optimization at each round. However, a scheduling ratio of 0.5 is impractical in real-world scenarios, as it incurs excessive communication bandwidth consumption and computational burdens on the server. Thus, we adopt a scheduling ratio of 0.2 for experiments in Section 6.1 to balance performance and practicality.

**Impact of $\eta$ and $\lambda$**: A small global learning rate $\eta$ leads to extremely slow convergence, requiring significantly more iterations to reach or approach the optimal solution. Conversely, a large $\eta$ may cause the model to skip the optimal region in the parameter space due to overly large step sizes. A small $\lambda$ can result in training fluctuations, while a large $\lambda$ weakens gradient accumulation, potentially impeding convergence.

**Bandwidth of kernel in KDE**: The influence of the bandwidth in KDE is relatively minor. In the context of our experiments, across different datasets and model configurations, varying the KDE bandwidth within a reasonable range did not lead to substantial changes in the model's performance metrics.

Table 5: Ablation studies on the impact of the number of labels per client, client scheduling ratio per communication round, $\eta$, $\lambda$ and KDE bandwidths on prediction performance, conducted with 100 clients on MNIST and FMNIST.

| MNIST | | FMNIST | |
|---|---|---|---|
| **NUMBER OF LABELS PER CLIENT** | | | |
| LABELS | ACC(%) | LABELS | ACC(%) |
| 2 | **99.26 ± 0.01** | 2 | **99.16 ± 0.06** |
| 5 | 97.23 ± 0.01 | 5 | 91.65 ± 0.02 |
| 10 | 95.37 ± 0.08 | 10 | 83.67 ± 0.07 |
| **CLIENT SCHEDULING RATIO PER COMMUNICATION ROUND** | | | |
| SCHEDULING RATIO | ACC(%) | SCHEDULING RATIO | ACC(%) |
| 0.1 | 96.78 ± 0.02 | 0.1 | 90.77 ± 0.04 |
| 0.2 | 97.57 ± 0.02 | 0.2 | 91.65 ± 0.02 |
| 0.5 | **97.71 ± 0.03** | 0.5 | **91.73 ± 0.01** |
| **GLOBAL LEARNING RATE OF ADAGRAD IN SVGD ($\eta$)** | | | |
| $\eta$ | ACC(%) | $\eta$ | ACC(%) |
| 0.01 | **97.23 ± 0.01** | 0.002 | 91.55 ± 0.02 |
| 0.02 | 96.53 ± 0.01 | 0.003 | 91.63 ± 0.02 |
| 0.03 | 95.59 ± 0.08 | 0.004 | **91.65 ± 0.02** |
| **$\lambda$ OF ADAGRAD FOR SVGD** | | | |
| $\lambda$ | ACC(%) | $\lambda$ | ACC(%) |
| $10^{-7}$ | 97.22 ± 0.01 | $10^{-7}$ | 91.64 ± 0.01 |
| $10^{-8}$ | **97.26 ± 0.02** | $10^{-8}$ | **91.67 ± 0.01** |
| $10^{-9}$ | 97.24 ± 0.01 | $10^{-9}$ | 91.66 ± 0.01 |
| $10^{-10}$ | 97.23 ± 0.01 | $10^{-10}$ | 91.65 ± 0.00 |
| **BANDWIDTH OF KDE** | | | |
| BANDWIDTH | ACC(%) | BANDWIDTH | ACC(%) |
| 0.30 | 97.25 ± 0.02 | 0.30 | 91.67 ± 0.01 |
| 0.55 | **97.26 ± 0.02** | 0.55 | **91.67 ± 0.01** |
| 0.70 | 97.24 ± 0.01 | 0.70 | 91.66 ± 0.01 |

### F.5 Communication Cost

First, we examine the relationship between the number of particles and communication cost. As stated in Section 4.2, with the increase in the number of particles $N$, the empirical distribution of particles asymptotically converges to the optimal variational distribution in the corresponding RKHS. In Table 6, we report the relationships among the number of particles, the size of uploaded data, and the prediction accuracy. Here, the unit of data size is megabyte (MB). Increasing the number of particles leads to a slight improvement in accuracy but imposes a communication burden. Therefore, 10 particles can strike a balance between communication volume and performance.

Table 6: Ablation study on the impact of particle count on communication overhead and prediction accuracy, conducted with 100 clients on MNIST and CIFAR-10.

| DATASET | NUMBER OF PARTICLES | COMM.(M) | ACC(%↑) |
|---------|----------------------|----------|---------|
| MNIST   | 5  | 1.52 MB  | $96.63 \pm 0.02$ |
|         | 10 | 3.03 MB  | $96.71 \pm 0.03$ |
|         | 20 | 6.07 MB  | $96.82 \pm 0.01$ |
|         | 50 | 15.17 MB | $\mathbf{96.86 \pm 0.01}$ |
| CIFAR-10 | 5  | 2.31 MB  | $70.71 \pm 0.10$ |
|          | 10 | 4.62 MB  | $72.00 \pm 0.14$ |
|          | 20 | 9.25 MB  | $72.14 \pm 0.09$ |
|          | 50 | 23.11 MB | $\mathbf{72.15 \pm 0.01}$ |

While using SVGD to approximate the posterior introduces moderate communication overhead due to particle updates, we also consider replacing Bayesian layers with standard frequentist layers to reduce costs. With 5 labels per client, we validate this trade-off using a 5-layer CNN for CIFAR-10 prediction, analyzing how the number of Bayesian layers affects communication cost, test accuracy, and ECE. When the number of Bayesian layers is reduced to 0, the method effectively reduces to FedAvg—a common non-Bayesian federated learning baseline—providing a direct comparison point.

Table 7: Impact of number of Bayesian layers on communication cost, test Accuracy, and ECE.

| THE NUMBER OF BAYESIAN LAYERS | COMM.(M) | ACC(%↑) | ECE(↓) |
|-------------------------------|----------|---------|--------|
| 5 | 23.99 MB | $\mathbf{76.85 \pm 0.02}$ | $\mathbf{0.0376}$ |
| 4 | 16.08 MB | $76.12 \pm 0.02$ | $0.0582$ |
| 1 | 2.40 MB  | $68.84 \pm 0.15$ | $0.0853$ |
| 0 | 2.38 MB  | $62.53 \pm 0.44$ | $0.1256$ |

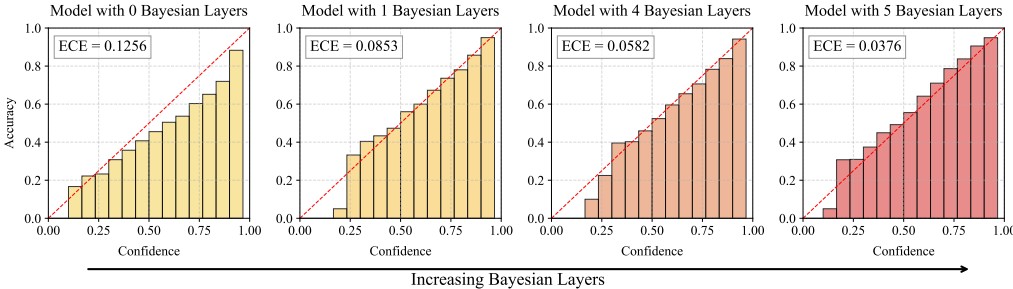

Figure 8: Reliability diagrams for different numbers of Bayesian layers.

As shown in Table 7 and Figure 8, decreasing the number of Bayesian layers lowers communication volume but leads to a noticeable drop in accuracy and an increase in ECE. This indicates that although Bayesian layers impose higher communication costs, they are critical for maintaining precise uncertainty quantification and prediction reliability. Specifically, our method's low ECE—even at higher communication costs—highlights its suitability for high-risk scenarios where accurate

uncertainty assessment is paramount, justifying the additional overhead. This trade-off underscores the value of our approach in applications requiring both performance and rigorous uncertainty quantification.

### F.6  Additional Experiments

To further verify the scalability of our method on complex datasets and large-scale networks, we also conduct experiments on Tiny-ImageNet (a reduced version of ImageNet) using the ResNet18 model. Specifically, the experimental setup includes 50 clients (each with 20 distinct classes to simulate non-iid data distribution) and runs for 100 communication rounds. These supplementary experiments further demonstrate the superiority of our method in such complex data scenarios and advanced network setups.

Table 8: Test accuracy (% ± SEM) over 50 clients on Tiny-ImageNet. Best results are bolded.

| DATASET | METHOD | ACC(%↑) | ECE(↓) |
|---------|--------|---------|--------|
| | FEDAVG | $27.14 \pm 0.12$ | 0.3170 |
| | FEDPER | $35.43 \pm 0.19$ | 0.2890 |
| | FEDPROX | $24.97 \pm 0.25$ | 0.2515 |
| TINY-IMAGENET | SCAFFOLD | $31.37 \pm 0.22$ | 0.0698 |
| | pFEDME | $36.24 \pm 0.14$ | 0.0631 |
| | PERAVG | $34.86 \pm 0.20$ | 0.1774 |
| | OURS | $\mathbf{37.91 \pm 0.07}$ | **0.0421** |

## G  Limitations and Future Work

FedWBA's key limitation stems from the computational complexity of SVGD for nonparametric posterior approximation at clients. While a fixed particle count balances efficiency, complexity rises significantly with higher-dimensional models. To address this, future work could explore lightweight Bayesian inference techniques, such as sparse particle approximations, to reduce computational overhead while retaining nonparametric flexibility.

