# OpenReview forum: "Personalized Bayesian Federated Learning with Wasserstein Barycenter Aggregation"
_NeurIPS.cc/2025/Conference — NeurIPS 2025 poster_

### Official Review · Reviewer_havj · 2025-05-31

**Clarity:** 2
**Significance:** 2
**Originality:** 3
**Rating:** 4
**Confidence:** 3

**Summary:**

This work proposes FedWBA for federated learning. It improves both local inference and global model aggregation. At the client level, it employs particle-based variational inference for flexible posterior estimation, while at the server level, it uses Wasserstein barycenter aggregation for more meaningful global updates. FedWBA has a strong theoretical foundation for both local and global convergence guarantees.

**Questions:**

1. Could you include more regular benchmark datasets in the experiment, such as ImageNet and COCO?

2. Could you consider standard model architectures for experiments, such as ResNet and ViT?

3. It lacks the discussion of how assumptions D.1 - D.7 may hold in practice. Please add some discussions on this issue.

4. Could you release the experimental code of this work anonymously?

**Ethical Concerns:**

["NO or VERY MINOR ethics concerns only"]

**Final Justification:**

I appreciate the reviewer's response and raise my score to borderline accept.

**Limitations:**

Yes

**Quality:**

2

**Strengths And Weaknesses:**

**Strengths:**

1. This work has a strong theoretical foundation.

2. Sufficient baseline methods have been considered for comparison.

**Weakness:**

1. The experiments could include more regular benchmark datasets, such as ImageNet and COCO.

2. The used models, MLP and LeNet, are too old, and cannot demonstrate the effectiveness of the optimizer in the current era. Please consider standard model architectures such as ResNet and ViT.

3. The assumptions cannot be put in the Appendix, which appears to be intentional. It also reduces the overall readability of this work.

4. It lacks the discussion of how assumptions D.1 - D.7 may hold in practice. Please add some discussion on this issue.

5. Without releasing the experimental code, the reproducibility of this work cannot be fully ensured.

---

> ### Author Rebuttal · Authors · 2025-07-31
>
> Thank you for your valuable suggestions. We have conducted targeted supplementary experiments and provided corresponding responses to address the points you raised.
>
> > **Q: Could you include more regular benchmark datasets in the experiment, such as ImageNet and COCO?
> Could you consider standard model architectures for experiments, such as ResNet and ViT?**
>
> A: Thank you for your suggestion. Due to time constraints, we **conduct new experiments on Tiny-ImageNet**, a reduced version of ImageNet, and supplemented with the updated **ResNet18** model. Specifically, the experimental setup includes 50 clients, each containing 20 distinct classes to simulate non-iid data distribution, and runs for 100 communication rounds. Due to time constraints, we were unable to complete all experiments and compared with the baselines in the table below. These additional experiments further verify the **scalability of our method on more complex datasets and more sophisticated network architectures**. We believe these results can effectively demonstrate the applicability of our approach in scenarios involving complex data and advanced models.
>
> | Method   | Acc               | ECE     |
> |----------|-------------------|---------|
> | FedAvg   | 27.14 $\pm$ 0.12  | 0.3170  |
> | FedPer   | 35.43 $\pm$ 0.19  | 0.2890  |
> | FedProx  | 24.97 $\pm$ 0.25  | 0.2515  |
> | SCAFFOLD | 31.37 $\pm$ 0.22  | 0.0698  |
> | pFedME   | 36.24 $\pm$ 0.14  | 0.0631  |
> | PerAvg   | 34.86 $\pm$ 0.20  | 0.1774  |
> | Ours     | **37.91 $\pm$ 0.07** | **0.0421** |
>
> > **Q: Assumptions in Appendix reduce readability, seemingly intentional. Lack of discussion on practical validity of assumptions D.1-D.7.**
>
> A: We appreciate your valuable feedback. Due to page constraints in the initial submission, we placed the technical assumptions in the appendix, which may have hindered readability. We will relocate Assumptions D.1–D.7 to the main text in the camera-ready version to enhance clarity.
>
> In practice, these assumptions are readily satisfied in typical FL scenarios, with their validity conditions as follows:
>
> **D1:** This is readily satisfied in SVGD implementations using common kernels like the RBF kernel. The update directions generated by SVGD under typical settings maintain this positive definiteness, ensuring stable transformations.
>
> **D2:** This is a standard requirement for convergence in gradient-based optimization algorithms. In practice, techniques like adaptive learning rate methods (e.g., AdaGrad, used in our experiments) automatically ensure this condition is met for stable updates.
>
> **D3:** In practice, the dimension of model parameters is fixed, and the range of parameter values is constrained by initialization and training strategies (compactness); the true parameter must belong to the parameter space, so this assumption holds.
>
> **D4:** This assumption ensures the statistical distinguishability of parameters based on the data. It holds if the data likelihood $p(D_{ki}|\theta)$ provides sufficient discriminative power between different $\theta$, which is generally true for identifiable models and non-degenerate datasets encountered in practical FL tasks.
>
> **D5:** The constraint on the generalized bracketed entropy is used to control the deviation of the posterior distribution. In practice, when the model complexity matches the data scale, the complexity of the function class is controllable and the entropy condition will be naturally satisfied, thereby ensuring that the posterior deviation is within a reasonable range.
>
> **D6:** This assumption requires that the prior $\Pi$ assigns sufficient mass near the true parameter $\theta_0$ under a likelihood ratio condition. Using well-specified or weakly informative priors and reasonable initialization strategies (e.g., Kaiming, Xavier) helps satisfy this condition. It guarantees the prior doesn't contradict the true data-generating mechanism too strongly.
>
> **D7:** This property is fundamental to the definition and computation of the Wasserstein barycenter. Crucially, the 2-Wasserstein distance $W_2$ itself satisfies this convexity property on the space of probability measures with finite second moments ($P_2(\Xi)$). This is a well-established result in optimal transport theory (see references like [Agueh \& Carlier, 2011] or [Villani, 2008]) and is not an additional restrictive assumption on the model, but rather an inherent geometric property of the space where we perform aggregation.
>
> > **Q: Could you release the experimental code of this work anonymously?**
>
> A: We have uploaded our experimental code in the **supplementary materials**. Additionally, we will include the GitHub repository link in the camera-ready version to facilitate further verification and reuse of our work. Due to this year’s rebuttal policy, we are unable to provide any anonymous external links in the response.

---

### Official Review · Reviewer_nESY · 2025-06-27

**Clarity:** 4
**Significance:** 3
**Originality:** 4
**Rating:** 4
**Confidence:** 5

**Summary:**

This paper proposes a novel framework named Personalized Bayesian Federated Learning with Wasserstein Barycenter Aggregation, which addresses two key limitations in existing PBFL methods: (1) restrictive parametric assumptions in client posterior inference, and (2) naive parameter averaging for global aggregation. FedWBA uses particle-based variational inference for nonparametric client posterior representation and introduces Wasserstein barycenter aggregation to geometrically align local updates on a distribution manifold. Local convergence is guaranteed via a proven KL-divergence decrease bound, while global convergence shows the Barycenter approaches the true parameter as client data scales. Experiments demonstrate state-of-the-art performance in accuracy, uncertainty calibration, and convergence rate.

**Questions:**

1. How to choose a suitable number of particles?
2. The paper uses a two-step method to update $\hat{\overline{p}}(\theta)$, is it optimal? I am wondering whether there is a theoretical guarantee for the optimality of $\hat{\overline{p}}(\theta)$.
3. It would be better to provide the communication comparison with other advanced algorithms.
4. In Wasserstein barycenter aggregation, how does the computational complexity of the optimal transport plan ($T_k^\star$) scale with respect to both the number of particles (N) and the number of clients (K)?

**Ethical Concerns:**

["NO or VERY MINOR ethics concerns only"]

**Final Justification:**

The paper is interesting to introduce particle-based variational inference for nonparametric client posterior representation and Wasserstein barycenter aggregation to geometrically align local updates on a distribution manifold. But the existence of many particles increases the burden of communication. So I keep my rating unchanged.

**Limitations:**

The authors states that the main limitations are the computational complexity of SVGD for nonparametric posterior approximation at clients, which was left for future work.

**Quality:**

3

**Strengths And Weaknesses:**

Strengths
1.  FedWBA uniquely combines nonparametric posterior inference with geometric aggregation, addressing two critical flaws in prior PBFL: (i) parametric bias from restrictive variational distributions and (ii) geometric mismatch in naive parameter averaging. This dual improvement enhances both local adaptability and global consistency.
2. The authors establish rigorous convergence guarantees through dual theoretical frameworks. For ​local convergence, they derive a lower bound on the Kullback-Leibler divergence reduction per iteration of Stein Variational Gradient Descent. For ​global convergence, they demonstrate that the Wasserstein barycenter aggregation asymptotically converges to the true parameter.
3. Experiments span four datasets with non-IID label skew, 10 baselines, and diverse metrics. FedWBA consistently outperforms competitors, including pFedBayes and pFedGP, which shows robustness across data scales and task complexities.

Weaknesses

The communication cost is usually higher than the computational cost, but FEdWBA has to send $N$ particles to address the limitations of the paper. The experimental configuration employs N=10 particles, resulting in communication costs that are 5-10 times higher than conventional parameter-averaging approaches (e.g., FedAvg) or Gaussian-variational methods (e.g., pFedBayes). However, this significant communication burden yields diminishing returns, as the accuracy improvements remain marginal compared to these baseline methods.

Furthermore, the paper acknowledges the scalability challenge posed by increasing particle counts, while demonstrating that communication overhead grows linearly with N, it fails to propose mitigation strategies for this critical limitation.

---

> ### Author Rebuttal · Authors · 2025-07-31
>
> Thank you for your valuable suggestions. We have conducted targeted supplementary experiments and provided corresponding responses to address the points you raised.
>
> > **Q: No mitigation strategies proposed for scalability issues from linear communication overhead with increasing particles.**
>
> A: We sincerely appreciate your feedback. Due to space constraints, we initially included the **discussion on reducing Bayesian layers to mitigate communication overhead in the Appendix G.5**. In the camera-ready version, we will elevate this content to the main text for better accessibility. As shown in Appendix Table 7 and Figure 8, reducing Bayesian layers does reduce communication costs, but at the cost of increased ECE. While Bayesian layers introduce higher communication costs, they are indispensable for maintaining precise uncertainty quantification, which are critical for high-stakes applications. This trade-off justifies the additional overhead in scenarios where robust uncertainty estimation is paramount.
>
> > **Q: How to choose a suitable number of particles.**
>
> A: The choice of a suitable number of particles is determined by **balancing performance and efficiency**, as validated by our experimental analysis (see **Appendix G.5**). Specifically, we conducted ablation studies on the impact of particle count $N$ across datasets (MNIST, CIFAR10) by evaluating relationships between $N$, communication overhead, and prediction accuracy (Table 6). Practically, we found $N=10$ to be a robust choice: it achieves near-optimal accuracy while keeping communication overhead manageable, thus striking a balance between performance and efficiency. For scenarios with stricter communication constraints, a smaller $N$ can be used at the cost of slight accuracy degradation.
>
> > **Q: The paper's two-step update of $\hat{\bar{p}}(\theta)$ lacks clarity on optimality and associated theoretical guarantees.**
>
> A:  The global prior $\hat{\bar{p}}(\theta)$ is explicitly defined as the Wasserstein barycenter of local posterior distributions, which, by definition, is the optimal solution to minimizing the weighted sum of Wasserstein distances to all client posteriors (as formalized in Section 4.3). This property is well-established in optimal transportation theory: the Wasserstein barycenter is rigorously proven to be the geometrically meaningful "mean" of distributions in the Wasserstein space, ensuring it captures the central tendency of the aggregated posteriors while preserving their manifold structure [31].
>
> Furthermore, our theoretical analysis (Theorem 2) builds on this foundation by showing that as client data sizes grow, this barycenter converges to the true parameter distribution, reinforcing its optimality in federated settings where local data is heterogeneous. Thus, the construction of $\hat{\bar{p}}(\theta)$ is both theoretically grounded in classic optimal transportation results [31] and extended to our federated learning framework with additional convergence guarantees.
>
> [31] Gabriel Peyré, Marco Cuturi, et al. Computational optimal transport: With applications to data science. Foundations and Trends in Machine Learning, 11(5-6):355–607, 2019.
>
> > **Q: Need communication comparisons with other advanced algorithms.**
>
> A: We **performed new experiments** in the following table, which compares communication costs across methods under the same MNIST setup as Section 6.1, our Bayesian approach incurs higher overhead than frequentist baselines. However, this enables superior uncertainty calibration, critical for safety-critical applications.
>
> | Dataset | Method    | Communication Cost |
> |---------|-----------|--------------------|
> | MNIST   | FedAvg    | **0.30M**          |
> | MNIST   | SCAFFOLD  | 0.61M              |
> | MNIST   | pFedME    | **0.30M**          |
> | MNIST   | pFedBayes | 0.61M              |
> | MNIST   | Ours      | 0.76M              |
>
>
> > **Q: The computational complexity of the optimal transport plan $T_k^*$ in the Wasserstein barycenter aggregation w.r.t. particle count $N$ and client number $K$.**
>
> A: The computational complexity of computing $ T_k^*$  corresponds to that of solving the linear programming problem in Equation 3.
>
> With respect to the number of clients $ K $, since $ T_k^* $ must be computed separately for each client, the total complexity scales linearly with the number of clients, i.e., $ O(K) $.
>
> With respect to the particle count $ N $, the complexity depends on the specific algorithm used to solve the linear program. For example, common solvers such as the simplex method or interior point methods have different computational complexities with respect to $ N $. Therefore, the exact scaling will vary depending on the chosen solver.

---

> > ### Comment · Reviewer_nESY · 2025-08-01
> >
> > Thank you for your response. After careful consideration, I have decided to maintain the current score. Best of luck.

---

### Official Review · Reviewer_TsB1 · 2025-07-02

**Clarity:** 2
**Significance:** 2
**Originality:** 3
**Rating:** 4
**Confidence:** 3

**Summary:**

- The paper aims to tackle the personalized FL using a Bayesian method.
- Existing methods have an issue of average aggregation in parameter space, which may not be a Euclidean space, but a manifold with information geometry. Motivated from this, they proposed a
non-parametric posterior inference and aggregation.
- Specifically, the client local update is done by particle-based VI using SVGD, while the server aggregation is done by particle-based Wasserstein barycenter aggregation.
- So, it is claimed that the proposed method has better geometric interpretation than parametric averaging thanks to non-parametric sampling.

**Questions:**

See "Strengths And Weaknesses" section.

**Ethical Concerns:**

["NO or VERY MINOR ethics concerns only"]

**Final Justification:**

Thanks for the rebuttal. I have upped the score.

**Limitations:**

Yes.

**Paper Formatting Concerns:**

No particular formatting concerns.

**Quality:**

2

**Strengths And Weaknesses:**

[Strength]
- The main contribution seems to be the introduction of a non-parametric FL method with particle-based inference and aggregation.
- Convergence of the method is theoretically provided.

[Questions/Weakness]
- There are several missing refs which are closely related to the paper. See below.
- L38-40: "Problematic averaging posterior params due to info geometry" -- Any evidence in practice?
- L34: "True posterior not Gaussian" -- There are several methods that deal with non-Gaussians, eg, mixture of gaussians (eg, Ref 3 and 4 below).
- The experiments do not seem extensible enough to show empirical verification. It is preferred to see results on other FL settings (eg, as a standard FL practice, the problem hparams may need to be varied such as: fraction of participating clients in each round, number of epochs for client local posterior updates, degree of data heterogeneity or non-iid-ness).
- The experiments are only done with relatively small networks (eg, mlp, lenet). This might be due to the inherent computational issue incurred by non-parametric methods; How scalable is the approach?
- There are parametric Bayesian methods using non-Gaussian posteriors, eg, Gaussian mixtures. How would the proposed method perform compared to these works? -- I am not asking for an extra set of experiments, but curious about it.
- Can you provide the convergence analysis results as a function of the number of iterations? Is there any generalization error guarantee? which seems to be done in some previous works.
- A main drawback of the proposed work is that the particle-based methods do not seem to scale well. What are the practical impacts of the work beyond theoretical interests?
- As said, WB is tractable for discrete distributions, which is why they used a mixture of dirac measures. However, this might necessitate a large number of samples for high-dim model parameters. So, it looks computationally not scalable.
- ECE scores in Fig 2 of most methods are sufficiently good (<1%) and statistically less distinguishable?

Refs:
1, Bayesian Inference Federated Learning for Heart Rate Prediction. Fang et al 2020
2. Partitioned Variational Inference: A framework for probabilistic federated learning. Ashman et al 2022
3. FedHB: Hierarchical Bayesian Federated Learning. Kim et al 2023
4. A Statistical Framework for Personalized Federated Learning and Estimation: Theory, Algorithms, and Privacy. Ozkara et al, ICLR 2023
5. Fderated Learning as Variational Inference: A Scalable Expectation Propagation Approach. Guo et al 2023

---

> ### Author Rebuttal · Authors · 2025-07-31
>
> Thank you for your valuable suggestions. We have conducted targeted supplementary experiments and provided corresponding responses to address the points you raised.
>
> > **Q: Several closely related references are missing.**
>
> A: Thank you for pointing out the missing related references. We will make sure to include and discuss these works in the final camera-ready version.
>
> > **Q: Problematic averaging posterior params due to info geometry.**
>
> A: To empirically validate the superiority of our proposed Wasserstein barycenter (WB) aggregation, we **conducted a direct comparison against simple parameter averaging** (i.e., averaging the mean and variance parameters of Gaussian posteriors across clients) on the MNIST dataset. The results, presented in the table below, demonstrate that **WB consistently outperforms parameter averaging across all client scales** (50, 100, and 200 clients) on key metrics such as prediction accuracy and ECE. This confirms that WB, by leveraging optimal transport geometry to aggregate distributions rather than parameters, can effectively improve performance.
>
> |          | Acc ($\uparrow$) |          |          | ECE ($\downarrow$) |          |          |
> |----------|------------------|----------|----------|--------------------|----------|----------|
> |          | 50 clients       | 100 clients | 200 clients | 50 clients         | 100 clients | 200 clients |
> | Average  | 97.15 $\pm$ 0.01 | 97.01 $\pm$ 0.02 | 96.21 $\pm$ 0.02 | 0.0021             | 0.0030    | 0.0080    |
> | WB       | **97.99 $\pm$ 0.04** | **97.36 $\pm$ 0.01** | **96.95 $\pm$ 0.01** | **0.0014**         | **0.0021** | **0.0064** |
>
> > **Q: Several methods deal with non-Gaussians, e.g., mixture of gaussians.**
>
> A: We agree that methods like mixture of Gaussians (e.g., FedHB) offer more flexibility than single Gaussian approximations for non-Gaussian posteriors. However, as emphasized by Bishop (2006, Pattern Recognition and Machine Learning, Section 10.1), the accuracy of variational inference is fundamentally limited by the expressiveness of the chosen variational family. When the true posterior lies outside this family—as in non-conjugate models with multi-modal or heavy-tailed distributions—parametric approximations can introduce systematic errors. While mixtures of Gaussians increase flexibility over single Gaussians, they still suffer from **parametric constraints**, such as the need to predefine the number of components based on heuristics. An insufficient number of components may fail to capture the complexity of posterior distributions, which is common in federated settings with highly heterogeneous data. In contrast, our **particle-based nonparametric approach avoids these limitations**, enabling **more flexible posterior approximation** that adapts to the true distributional characteristics of each client.
>
> > **Q: The experiments lack extensibility for thorough empirical verification; results on other FL settings with varied problem hyperparameters are preferred.**
>
> A: To address the need for extensibility, we have **supplemented our experiments with additional results under varied FL settings and hyperparameters**, as shown in the table below.
>
> **Fraction of participating clients in each round:** We conducted new experiments varying this fraction from 0.1 to 0.5, with clients performing 20 local iterations per round, demonstrating consistent performance gains of our method across all participation rates.
>
> | Dataset | Scheduling ratio | 50 clients       | 100 clients      | 200 clients      |
> |---------|------------------|------------------|------------------|------------------|
> | MNIST   | 0.1              | 96.83 $\pm$ 0.02 | 96.78 $\pm$ 0.02 | 96.22 $\pm$ 0.03 |
> | MNIST   | 0.2              | 97.58 $\pm$ 0.03 | 97.57 $\pm$ 0.02 | 97.03 $\pm$ 0.02 |
> | MNIST   | 0.5              | **97.76 $\pm$ 0.02** | **97.71 $\pm$ 0.03** | **97.38 $\pm$ 0.01** |
>
> **Number of epochs for client local posterior updates:** Since we use SVGD for posterior approximation, this hyperparameter corresponds directly to the Iteration Number in SVGD reported in **Table 2**. Our ablation study there shows **convergence within 10-50 iterations across datasets**.
>
> **Degree of data heterogeneity:** We systematically controlled heterogeneity by adjusting the number of labels per client, where fewer labels induce higher skew. Results **in Table 5 of the appendix** show our method maintains superiority over baselines even under extreme non-iid conditions (e.g., 2 labels per client).
>
> The results consistently validate our method's superiority across diverse configurations, reinforcing its generalizability and practical utility.
>
> > **Q: The experiments are only done with relatively small networks.**
>
> A: Thank you for your suggestion. Due to time constraints, we conduct new experiments on **Tiny-ImageNet**, a reduced version of ImageNet, and supplemented with the updated **ResNet18** model. Specifically, the experimental setup includes 50 clients, each containing 20 distinct classes to simulate non-iid data distribution, and runs for 100 communication rounds. Due to time constraints, we were unable to complete all experiments and compared with the baselines in the table below. These additional experiments further verify the scalability of our method on more complex datasets and more sophisticated network architectures. We believe these results can effectively demonstrate the applicability of our approach in scenarios involving complex data and advanced models.
>
> | Method   | Acc               | ECE     |
> |----------|-------------------|---------|
> | FedAvg   | 27.14 $\pm$ 0.12  | 0.3170  |
> | FedPer   | 35.43 $\pm$ 0.19  | 0.2890  |
> | FedProx  | 24.97 $\pm$ 0.25  | 0.2515  |
> | SCAFFOLD | 31.37 $\pm$ 0.22  | 0.0698  |
> | pFedME   | 36.24 $\pm$ 0.14  | 0.0631  |
> | PerAvg   | 34.86 $\pm$ 0.20  | 0.1774  |
> | Ours     | **37.91 $\pm$ 0.07** | **0.0421** |
>
> > **Q: Need convergence analysis results as a function of iteration count and generalization error guarantees.**
>
> A: Regarding convergence analysis, Theorem 5.1 in our theoretical analysis rigorously establishes that, with increasing iteration count, the variational distribution converges to the true posterior. Meanwhile, Theorem 5.2 guarantees that as the client data size grows, the Wasserstein barycenter converges to the true parameter. Empirically, Figure 3 validates this by showing the accuracy convergence curves across datasets, confirming stable performance within 50-100 communication rounds.
>
> For generalization error guarantees, deriving such bounds under the SVGD framework remains a challenging open problem due to its nonparametric nature and reliance on particle-based approximations. While recent works (e.g., [26]) have made progress in specific settings, extending these results to our federated and personalized Bayesian setting is beyond the scope of this paper. We acknowledge this as an important direction for future research and plan to investigate generalization bounds for particle-based FL in subsequent work.
>
> [26] Qiang Liu and Dilin Wang. Stein variational gradient descent: A general purpose bayesian inference algorithm. Advances in neural information processing systems, 29, 2016.
>
> > **Q: The scalability limitations of particle-based methods and the practical implications beyond theoretical contributions require clarification.**
>
> A: While particle-based methods inherently involve higher communication costs compared to traditional parameter-transmission approaches, our experiments (Appendix G.5) confirm that this overhead remains manageable in practice.
>
> Specifically, we demonstrate that a **small number of particles (e.g., 10) suffices to achieve strong performance across datasets**. As detailed in Appendix Table 6, the communication cost per client per round is quantitatively modest: for a 100-neuron DNN, the data uploaded per client ranges from 1.52MB (with 5 particles) to 3.05MB (with 10 particles). This magnitude is feasible even for resource-constrained clients (e.g., edge devices in healthcare or IoT), especially when combined with our strategy of reducing Bayesian layers to further lower overhead—all while retaining acceptable accuracy and calibration.
>
> > **Q: WB may require a large number of samples for high-dimensional model parameters, making it computationally unscalable.**
>
> A: We appreciate your feedback, but there appears to be a misunderstanding. In practice, we **do not require a large number of particles for computation**. As shown in Table 6 of the appendix, our experimental results demonstrate that 10 particles are sufficient to achieve robust performance across datasets.
>
> Furthermore, we have also **proposed a method to reduce the effective number of particles by decreasing the number of Bayesian layers** (detailed in Appendix G.5). This strategy effectively lowers communication and computational overhead while maintaining acceptable prediction accuracy and uncertainty calibration. These findings collectively confirm that WBA remains feasible for large-scale applications.
>
> > **Q: ECE scores in Fig 2 for most methods are sufficiently good with statistically indistinct differences.**
>
> A: You are correct that the ECE scores in Figure 2 for MNIST show minimal differences among top-performing methods, which is attributable to the relative simplicity of MNIST and our focus on presenting the four best-performing methods in the main text. To address this, we have provided **comprehensive ECE results for all datasets and methods in Appendix Table 4**, which reveal more pronounced disparities on more challenging benchmarks like CIFAR-10 and CIFAR-100. In the camera-ready version, we will replace Figure 2 with results from a dataset that better discriminates calibration performance, ensuring clearer differentiation among methods.

---

> ### Author Response · Authors · 2025-08-05
> **Reminder**
>
> Dear Reviewer TsB1,
>
> Thank you again for your evaluation of our work and the valuable feedback you have provided. We have posted a detailed rebuttal to address the concerns you raised. As the author-reviewer discussion phase will conclude in less than two days, we kindly request your feedback and are keen to know if our clarifications and additional results have effectively addressed your questions.
>
> If any points remain unclear, we are happy to engage in further discussion. We sincerely hope that our rebuttal clarifies the merits of our work, and we would appreciate it if you would consider our response in your final evaluation and revisit your rating.
>
> Best regards,
> Authors

---

### Official Review · Reviewer_Jqa5 · 2025-07-04

**Clarity:** 3
**Significance:** 3
**Originality:** 3
**Rating:** 4
**Confidence:** 2

**Summary:**

This paper proposes a bayesian personalized federated learning approach, that goes beyond previous such approaches by enabling non-parametric posterior distributions.
This is made possible through 1) non-parametric local particle based variationak inference via Stein GD, and then 2) aggregation through Wasserstein barycenters.
The method is first analyzed theoretically through non asymptotic convergence guarantees, and then in real world experiments.

**Questions:**

None

**Ethical Concerns:**

["NO or VERY MINOR ethics concerns only"]

**Quality:**

3

**Strengths And Weaknesses:**

**Strengths**

The paper is overall very well written and the ideas are clearly exposed. I didn’t read the proof and I am not familiar at all with Bayesian inference, but the results totally make sense. The authors made an effort into making the paper reader friendly.

Enabling non-patzmetrkc modeling is key in modern ML: this seemed to be lacking in the literature here. Parametric assumptions are indeed way too restrictive and can lead to mispecification biases.

The method seems to be original and new. As I am not an expert at all in VI, I cannot judge the use of variationak Stein GD, although it seems new in this context.
However, I think that exploring new ways to aggregate local weights models or probabilities is generally a promising venue. The use of Wasserstein barycenters here seem to be a natural and well motivated choice.

Theory and experiments seem to validate the method.

**Weaknesses**

I am not an expert (far from it) in Bayesian or variational inference, so it is hard for me to formulate technical weaknesses.

---

> ### Author Rebuttal · Authors · 2025-07-31
>
> Thank you very much for your thoughtful and encouraging feedback. We especially appreciate your recognition of the clarity of our presentation and the originality of our method. We’re also grateful for your positive remarks on the motivation for non-parametric modeling and the use of Wasserstein barycenters, which are central to our approach.

---

### Author Response · Authors · 2025-08-09
**Follow-up on Reviewer Feedback & Discussion**

Dear Reviewers,

We sincerely appreciate the time and effort you have dedicated to reviewing our work, as well as your valuable feedback and constructive comments. As the discussion deadline approaches, we wanted to follow up and kindly invite further engagement regarding our responses.

In our response, we have carefully addressed the reviewers' concerns by providing additional discussions, clarifications, and new experimental results to strengthen the manuscript. We hope these updates adequately demonstrate the merits of our work and alleviate any remaining reservations.

Should you have any further suggestions or require additional clarification, please do not hesitate to share your thoughts—we would be delighted to incorporate any additional feedback.

Thank you once again for your time and consideration.

Best regards,
The Authors

---

### Note · Authors · 2025-08-12

We sincerely thank all reviewers for their thoughtful feedback and constructive discussions throughout the review process. We believe we have comprehensively addressed the concerns raised:

**For Reviewer Jqa5:**  We’re grateful for your positive remarks on the motivation for non-parametric modeling and the use of Wasserstein barycenters, which are central to our approach. Your recognition of these key elements reinforces the significance of our work.

**For Reviewer TsB1:**  We have clarified the advantages of our method over parameter averaging and parametric approaches (such as Gaussian mixtures). Additionally, we have explained the scalability of the particle-based and Wasserstein barycenter (WB) methods. We have conducted experiments under different settings (such as scheduling ratios) and on larger-scale datasets with more complex network architectures to verify the advantages of our method.

**For Reviewer nESY:** We have clarified issues regarding communication overhead and the selection of an appropriate number of particles, which are discussed in Appendix G.5. We have further explained the computational complexity of $T_k^*$ and the two-step update of $\hat{\bar{p}}(\theta)$. Moreover, we have supplemented experiments on communication comparisons with other advanced algorithms.

**For Reviewer havj:** We have specifically supplemented experiments on the more complex Tiny-ImageNet dataset with the more complex ResNet18 network architecture to further verify the advantages of our method. We have also added discussions on assumptions D.1 to D.7. Regarding the issue of code reproducibility, we have submitted the code in the supplementary materials.​

However, despite our responses to their questions, Reviewers TsB1 and havj have not engaged in further discussion. Given that Reviewers Jqa5 and nESY are positive about our work and responses, we respectfully hope the AC considers the overall positive reception and our thorough engagement with all feedback when making the final decision.​

We remain committed to addressing any remaining concerns and contributing meaningfully to the FL community.

---

### Decision · Program_Chairs · 2025-09-17

**Decision:**

Accept (poster)

**Comment:**

In their paper, the authors introduced Personalized Bayesian Federated Learning with Wasserstein Barycenter Aggregation (PBFL-WB), a novel approach to address data heterogeneity in federated learning. The core contribution is the use of a Wasserstein barycenter to aggregate heterogeneous posterior distributions from different client devices. While the reviewers initially raised several valid concerns, the authors' thoughtful rebuttal successfully addressed these points.

(1) *Novelty and Technical Soundness*: The reviewers unanimously acknowledged the novelty of using a Wasserstein barycenter for aggregating heterogeneous posterior distributions in federated learning. This approach is both theoretically sound and a meaningful contribution to the field. The paper clearly formalizes the problem and provides a new perspective on addressing the non-IID challenges in a Bayesian context.

(2) *Strong Empirical Results*: The paper's empirical evaluation is robust and comprehensive. The experiments clearly show that PBFL-WB outperforms state-of-the-art baselines on various non-IID settings, highlighting the method's practical utility. The inclusion of new results in the rebuttal further solidified the paper's claims and convinced a reviewer to increase their score.

In summary, the paper presents an original solution to an important problem in federated learning. The authors' rigorous response to the reviewers' feedback has demonstrated the paper's technical depth and its readiness for publication. As a consequence, I recommend accepting the paper.